# Compliant-Control-Based Assisted Walking with Mobile Manipulator

**DOI:** 10.3390/biomimetics9020104

**Published:** 2024-02-09

**Authors:** Weihua Li, Pengpeng Li, Lei Jin, Rongrong Xu, Junlong Guo, Jianfeng Wang

**Affiliations:** 1School of Automotive Engineering, Harbin Institute of Technology-Weihai, Weihai 264201, China; lipengpeng5033@163.com (P.L.); 18132835162@163.com (L.J.); xurongrong97@163.com (R.X.); wjfeee123@163.com (J.W.); 2Yangtze River Delta HIT Robot Technology Research Institute, Wuhu 241060, China; 3State Key Laboratory of Robotics and System, Harbin Institute of Technology, Harbin 150001, China

**Keywords:** compliant control, assisted walking, fuzzy theory, mobile manipulator

## Abstract

In this paper, a new approach involving the use of a mobile manipulator to assist humans with mobility impairments to walk is proposed. First, in order to achieve flexible interaction between humans and mobile manipulators, we propose a variable admittance controller that can adaptively regulate the virtual mass and damping parameters based on the interaction forces and the human motion intention predicted using the fuzzy theory. Moreover, a feedforward velocity compensator based on a designed state observer is proposed to decrease the inertia resistance of the manipulator, effectively enhancing the compliance of the human–robot interaction. Then, the configuration of the mobile manipulator is optimized based on a null-space approach by considering the singularity, force capacity, and deformation induced by gravity. Finally, the proposed assisted walking approach for the mobile manipulator is implemented using the human–robot interaction controller and the null-space controller. The validity of the proposed controllers and the feasibility of assisted human walking are verified by conducting a set of tests involving different human volunteers.

## 1. Introduction

In recent years, the elderly population has globally increased rapidly, and there is a great need for assisted walking robots to help the elderly or disabled people walk. Among the multitude of devices and robots, especially walking assistants, wheelchairs have long served as a good option for this group of people. However, wheelchairs can lead to muscle weakness and joint stiffness, which can hamper the recovery of such people’s athletic abilities. Therefore, many scholars have developed and studied various types of devices for providing assistance to the elderly [1,2,3,4,5,6], such as powered lower-limb orthoses, wearable exoskeletons, intelligent wheelchairs, walkers, and assistive robots. Robot-assisted exercise training can provide sustainable intensive rehabilitation treatment to the elderly and help them recover their motion functions [7]. Traditional walking aids include the U-Step walking stabilizer, whose velocity can be adjusted by cranking or height changes. However, the use of such assistants increases the risk of falling [8]. Smart walkers can provide superior walking assistance, but several of these devices are extremely complex to use [4]. 

Mobile manipulators have been widely used in walking assistance robots, which generally consist of a mobile base and a manipulator. Herein, the main challenge is realizing cooperative motion control between the mobile manipulator and the human. Mobile manipulators have been used to assist humans with opening doors and carrying heavy objects [9,10,11]. To obtain the desired human–robot commands, several controllers have been proposed, including compliant control, human motion intention recognition, and redundant solutions.

In the task of assisted walking, the interaction force from the human hand should be decreased by using compliant control. This allows humans to interact effortlessly with the mobile manipulator.

In general, compliant control can be divided into active and passive compliance [12]. Active compliance can be further subdivided into impedance control, admittance control, and hybrid force/position control [13]. Passive compliance has a critical disadvantage in that it cannot be used to vary the compliance effect depending on the external environment, and it is effective only in a few simple applications [14]. 

Compared with passive compliance, with active compliance, it is easier to realize and modify the compliance effect of the robot. In manipulators, admittance controllers are widely used to achieve compliance control [15]. Thus far, active compliance has invariably been implemented by regulating the virtual damping, virtual stiffness [16], virtual mass, or simultaneously changing the virtual damping parameter and virtual mass. For example, Jiang et al. [17] proposed a variable admittance algorithm based on Fuzzy Sarsa (λ)-learning to regulate the virtual damping parameters. Kuang et al. [18] proposed a variable admittance controller based on fuzzy reference learning control by changing the virtual damping parameters in the admittance model to ensure that the manipulator moves according to the surgeon’s motion intention, thus exhibiting a certain level of compliance. However, these methods can provide good results only on specific tasks, and they have relatively strong limitations. In addition, radial basis function (RBF) neural networks have been widely used to adjust the damping parameters of admittance controllers [19]. Sharkawy et al. [20] proposed a method to modify only the virtual mass parameter through online training of a multilayer feedforward neural network. Thereafter, they proposed another method to simultaneously regulate the virtual mass and virtual damping by using a Jordan recurrent neural network [21].

In the task of assisted walking, it is also necessary to recognize the human motion intention to realize forward, backward, and steering operations, ensuring that mobile manipulators can effectively track human motion and provide assistance.

To identify human motion intention, video-based image recognition methods such as the OpenPose algorithm [22,23] are widely used to identify human hands and limbs. However, during the walking process, due to the camera view, it is usually difficult to obtain all the image information of the limbs.

In addition, sensor-based human behavior recognition methods [24] have emerged recently, and they can be subdivided into methods based on wearable and non-wearable sensors. The wearable sensors used in such methods include electroencephalography (EEG) collectors, electrooculography (EOG) collectors [25,26] and electromyography (EMG) collectors [27]. However, it is common for signals to be easily interfered with, difficult to measure, and the wearing process to be complex, which brings significant inconvenience to daily care.

Therefore, many scholars have studied intention-recognition methods based on non-wearable sensors. In [28], the use of high-order Markov chains to gauge human motion intentions was proposed. Tomislav et al. [29] proposed a method to predict human behavioral intentions by using a hidden Markov model. Palm et al. [30] proposed a collision-avoidance method based on fuzzy control rules and verified its effectiveness by conducting a simulation and an experiment. However, the identification of human motion intention during walking is fraught with several novel challenges, like how to accurately identify the intention of human movement without wearing any sensors in the contact interaction. In such cases, information pertaining to interaction forces and velocities effectively serves as the source data. Lecours et al. [31] proposed a variable admittance controller that can infer human intentions by using velocity and acceleration data. 

In the task of assisted walking, the mobile manipulator should also be able to move reasonably. Generally, a mobile manipulator is a system that possesses redundant degrees of freedom. For solving such redundancy, usually the extended Jacobi method [32] and the gradient projection method [33] are employed. Cetin et al. [34] proposed an extended Jacobi matrix formulation for a fixed laser or optic camera mounted on the first joint to accurately track the end-effector. Boundec et al. [35] employed the gradient projection method to realize obstacle avoidance. Under this method, they developed three objective functions to achieve obstacle avoidance, help the robot avoid exceeding its physical limits, and prevent the robot from colliding with its own base.

Therefore, we propose an approach by employing a mobile manipulator to assist human walking in this study, which involves having an experienced nurse help a patient walk. The mobile manipulator used in this work consists of a four-wheeled mobile robot and a six-DOF (degree of freedom) manipulator. Assisted walking based on mobile manipulators can provide sustainable, intensive rehabilitation for patients with mobility impairments. The main challenges of assisted walking depend on the flexibility of human–robot interaction, the configuration optimization of the mobile manipulator, and the smoothness and robustness of the human–robot collaborative controller. 

To implement compliant-control-based walking assistance by using a mobile manipulator, we propose a human motion intention recognition method and a variable admittance controller based on the fuzzy theory, which can accurately predict human motion intention. Simultaneously, we eliminate the inertial resistance of the manipulator by means of velocity feedforward compensation provided using a developed state observer, which can effectively enhance the compliance of the interaction between the human and the mobile manipulator. Furthermore, to solve the redundant degrees of freedom of the mobile manipulator, a null-space controller based on the gradient projection method is proposed. This controller can identify the optimal manipulator configuration. To guarantee that the position and posture of the end-effector remain unchanged during assisted walking, a position and posture controller for the end-effector is proposed based on the proportional control algorithm. The main contributions of this paper are as follows:A variable admittance controller is proposed to simultaneously adjust virtual mass and virtual damping based on the fuzzy theory, and a state-observer-based velocity compensator is also proposed.A null-space controller is presented to optimize the mobile manipulator’s configuration so that it can avoid singularities, enhance force capacity in the vertical direction, and optimize gravity-induced deformation.By using the aforementioned controllers, the mobile manipulator is able to successfully provide walking assistance to humans.

The remainder of this paper is structured as follows: In Section 2, the kinematics model of the mobile manipulator is established. In Section 3, a human–robot-compliant interaction method is proposed based on the fuzzy theory. In Section 4, we propose a method to optimize the configuration of the mobile manipulator based on a null-space controller. Section 5 describes a series of experiments that verify the effectiveness and correctness of the methods proposed herein. Our concluding remarks are presented in Section 6.

## 2. Kinematics Model of Mobile Manipulator

In this paper, a mobile manipulator consisting of a four-wheeled differential mobile robot and a 6-DOF manipulator is researched, as illustrated in Figure 1.

To coordinate the kinematic relationship between the mobile platform and the manipulator joints, we first develop the integrated kinematics of the mobile manipulator. The world coordinate system, mobile platform coordinate system, manipulator base coordinate system, and manipulator end-effector coordinate system are represented by Σ_w_, Σ_b_, Σ_0_, and Σ_ee_, as depicted in Figure 1.

Herein, the Denavit–Hartenberg (D–H) method is used for kinematic modeling of the mobile manipulator. ***q***_b_ = [*x*_b_ *y*_b_ *φ*]^T^ represents the position and orientation of the mobile base coordinate system in the world coordinate system, where *x*_b_, *y*_b_, and *φ* are respectively the *x*-coordinate, *y*-coordinate, and rotation angle of the mobile platform in the world coordinate system. ***q***_m_ = [*q*_1_ *q*_2_ *q*_3_ *q*_4_ *q*_5_ *q*_6_]^T^ is the joint angle vector of the manipulator, and *q*_j_ is the angle of the j-th (j = 1, 2, 3, 4, 5, 6) joint. Therefore, the transformation matrix from Σ_b_ to Σ_w_ can be obtained as follows:(1)Tbw=cosφ−sinφ0xbsinφcosφ0yb00100001.
(2)$$\left\{\begin{array}{c} u_{b}=\left[\begin{array}{l} v_{b}\\ \omega_{b} \end{array}\right]\\ {\dot{q}}_{b}=\left[\begin{array}{cc} \cos\varphi & 0\\ \sin\varphi & 0\\ 0 & 1 \end{array}\right]\left[\begin{array}{l} v_{b}\\ \omega_{b} \end{array}\right]=P\left(q_{b}\right)u_{b} \end{array}\right..$$
where ***P***(***q****_b_*) and ***u****_b_* are the kinematic constraint and velocity vector of the mobile base, respectively. *v*_b_ and *ω*_b_ are the longitudinal and heading angle speeds of the mobile base, respectively.

In addition, the transformation matrix from the end-effector to the manipulator base is defined as T60, and the transformation matrix from the manipulator base to the mobile platform is defined as T0b, which are both the constant matrixes. Then, the transformation matrix from the end-effector to the world coordinate system can be expressed as Tew=TbwT0bT60.

By using the matrix Tew, the Jacobian matrix of the mobile manipulator can be constructed, including the differential kinematic model of the mobile platform. Herein, we assume that the DOF of the manipulator in the task space is *r*, the DOF in the joint space is *m*, the DOF of the mobile robot is *n_b_*, and its degree of mobility under the nonholonomic constraints is *b*. The total DOF of the mobile manipulator is *n* = *n_b_* + *m*. Then, the kinematic model of the mobile manipulator can be expressed as follows:(3)$$\left\{\begin{array}{c} \dot{x}=J\left(q\right)\dot{q}=\left[\begin{array}{cc} J_{b}\left(q\right) & J_{m}\left(q_{m}\right) \end{array}\right]\left[\begin{array}{c} {\dot{q}}_{b}\\ {\dot{q}}_{m} \end{array}\right]=\left[\begin{array}{cc} J_{b}\left(q\right)P\left(q_{b}\right) & J_{m}\left(q_{m}\right) \end{array}\right]\left[\begin{array}{c} u_{b}\\ u_{m} \end{array}\right]=J_{u}\left(q\right)u\\ u=J_{u}{\left(q\right)}^{-1}\dot{x} \end{array}\right..$$
where r=m=6, $J\in R^{r\times n}$ is the Jacobian matrix of the mobile manipulator without considering the nonholonomic constraints; $J_{b}\in R^{r\times n_{b}}$ is the Jacobian matrix of the mobile platform, and it is related to the matrix in (1); $J_{m}(q_{m})\in R^{r\times m}$ is the Jacobian matrix of the manipulator; $\dot{x}\in R^{r}$ is the velocity vector of the end-effector; ${\dot{q}}_{b}\in R^{n_{b}}$ is the velocity vector of the mobile platform; $u_{m}\in R^{m}$ is the velocity vector of the manipulator joints; and $u_{m}={\dot{q}}_{m}$, $J_{u}\in R^{r\times (b+m)}$ is the Jacobian matrix of the mobile manipulator considering the nonholonomic constraints.

By using the mobile manipulator model (3), we can reasonably allocate ***u***_b_ and ***u***_m_.

## 3. Human–Robot Compliant Interaction Based on Fuzzy Control

To realize compliant interaction between the manipulator and the human user in the process of providing walking assistance, the admittance control method is employed in this study. A virtual “Inertia-Damper-Spring” model is established between the manipulator and the human hand. The target velocity, or angular velocity, of the end-effector is calculated based on the interaction force or torque exerted on the manipulator by the human hand. This method mainly realizes the compliant interaction between the mobile manipulator and the human hand in the horizontal direction and provides adequate support to the human in the vertical direction. 

Herein, we focus on the performance of force interaction between the human and the manipulator while ignoring the interaction torque and spring effect. The admittance model used in this study can be expressed as follows:(4)$$\left\{\begin{array}{c} M_{v}\ddot{x}+C_{v}\dot{x}=F_{h}\\ M_{v}=\mathrm{diag}\left\{\begin{array}{ccc} M_{x} & M_{y} & M_{z} \end{array}\right\}\\ C_{v}=\mathrm{diag}\left\{\begin{array}{ccc} C_{vx} & C_{vy} & C_{vz} \end{array}\right\} \end{array}\right..$$
where $M_{v}\in R^{3\times 3}$ is a virtual mass matrix; *M*_x_, *M*_y_, and *M*_z_ are virtual masses in the x, y, and z-directions, respectively; $C_{v}\in R^{3\times 3}$ is a virtual damping matrix; *C_v_*_x_, *C_v_*_y_, and *C_v_*_z_ denote virtual damping in the x, y, and z-directions, respectively; and $F_{h}\in R^{3\times 1}$ is the human–robot interaction force.

For the single-DOF admittance model, the transfer function between the end-effector velocity and the interaction force can be expressed as follows:(5)$$\left\{\begin{array}{c} H(s)=\frac{\dot{x}(s)}{F_{h}(s)}=\frac{1}{M_{v}s+C_{v}}=\frac{1/C_{v}}{\beta s+1}\\ \beta =M_{v}/C_{v} \end{array}\right..$$
where *β* is the ratio of virtual mass to virtual damping.

*C*_v_ and *β* affect the steady-state response and poles of the interaction system, thereby affecting the dynamic response characteristics of the system. If *β* is small, the response speed of the system will be faster, but the system will be prone to overshoot and vibration. By contrast, if *β* is large, the system will be more stable, but its response time will be longer.

In walking-assistance technology, when the mobile manipulator moves rapidly with the human, the interaction system needs a small damping coefficient and offers superior compliance. When the human user decelerates or stops moving, the interactive system needs a large damping coefficient to offer superior stability and accuracy. However, it is difficult to implement such a regulation by using the traditional admittance control scheme with fixed parameters, and for this reason, a variable-structure admittance controller is proposed herein.

In [17,36], the variable structure admittance controller only adjusts ***C***_v_ while keeping ***M***_v_ unchanged, resulting in the change of *β*. However, ***M***_v_ also affects the magnitude of ***F***_h_ in the process of acceleration and deceleration. To further improve the performance of the interaction system, we propose that ***M***_v_ should be varied with ***C***_v_. Therefore, according to [31], we define the virtual mass as a piecewise function of virtual damping, as expressed in the following equation: A schematic diagram of piecewise function (6) is shown in Figure 2.
(6)$$M_{vi}=\left\{\begin{array}{ll} \beta C_{vi} & C_{vi}\leq C_{0}\\ \beta \left(1-\varepsilon \left(1-e^{-\mu \left(C_{vi}-C_{0}\right)}\right)\right)C_{vi} & C_{vi}>C_{0} \end{array}\right..$$
where $C_{vi}$ is the virtual damping coefficient in the *i* direction, and *i* represents the x, y, and z-directions, which is a scalar. *β* is the ratio of steady-state virtual mass to virtual damping, and *β* > 0; *ε* is the adjustment factor of steady-state virtual mass to the virtual damping ratio, and it satisfies 0 < *ε* < 1; *μ* is the smoothness factor, and it satisfies μ > 0; *i* represents the x, y, and z-directions. *C_0_* represents the virtual damping coefficient in steady state, and *C*_0_
*=* 50 Ns/m in this paper.

To realize the adaptive variable structure admittance control, we adjust the virtual damping according to the end-effector velocity and the interaction force between the manipulator and the human. In view of the robustness of fuzzy control, we build a fuzzy function to describe the relationship between $C_{vi}$,$\;v_{hi},$ and *F_hi_*.
(7)$$\left\{\begin{array}{c} C_{vi}=f\left(v_{hi},F_{hi}\right),\;i=1,2,3\\ C_{vi}\in \left[\begin{array}{cc} C_{\min} & C_{\max} \end{array}\right] \end{array}\right..$$
where $f\left(v_{hi},F_{hi}\right)$ is a fuzzy function of $v_{hi}$ and *F_hi_*, which will be covered in the next section. *C*_max_ and *C*_min_ are, respectively, the upper and lower limits of the virtual damping adjustment, which corresponds to the upper and lower limits of fuzzy function values; *F_hi_* is the interaction force; and $v_{hi}$ is the end-effector velocity. In this paper, the minimum and maximum values of *C_vi_* are set to be *C*_min_ = 30 Ns/m and *C*_max_ = 70 Ns/m.

The above equation can be used to effectively regulate the relationship between the virtual mass and virtual damping to achieve superior human–robot interaction performance in the case of assisted walking. Then, the desired end-effector velocity ***v***_d_ is determined using the modified admittance controller. By (3), we can obtain the desired ***u***_b_ and ***u***_m_, and the movement of the mobile manipulator according to the human user’s motion intention during the process of walking assistance is completed simultaneously.

### 3.1. Prediction of Human Motion Intention Using Fuzzy Theory

In this section, the fuzzy theory is employed as it can well describe the complex relationship between the required damping coefficients and human–robot interaction motion and forces and has good robustness for the uncertainty of human–robot interaction. Based on a series of pretests, the interaction force and velocity of the end-effector can be used to represent the human motion intention, and the suitable damping is further analyzed to improve human comfort. Further, the fuzzy theory is employed to establish the relationship between virtual damping and human–robot interaction.

(a)When the interaction force ***F***_h_ or the end-effector velocity ***v***_h_ increases gradually from 0, it can be judged that the human user intends to start moving, and accordingly, the damping coefficient should be decreased gradually.(b)When the directions of ***F***_h_ and ***v***_h_ are the same and their magnitudes are large, it can be judged that the human user intends to accelerate significantly, and accordingly, a small damping coefficient is required.(c)When the directions of ***F***_h_ and ***v***_h_ are not the same but their magnitudes are large, it can be judged that the human user intends to decelerate significantly, and accordingly, a large damping coefficient is required.(d)When ***F***_h_ and ***v***_h_ are extremely small, it can be judged that the human user intends to maintain the current position or move with greater precision, and accordingly, a large damping coefficient is required.

Then, the virtual damping ***C***_v_ in (7) can be regulated based on the prediction results. Thus, the inputs to the proposed fuzzy system are ***F***_h_ and ***v***_h_, and its output is the virtual damping ***C***_v_. Assuming that the maximum and minimum values of ***F***_h_ in the three directions are identical as *F*_max_ and *F*_min_, respectively, and the maximum and minimum values of ***v***_h_ in the three directions are defined to be equal as *v*_max_ and *v*_min_, respectively, the domains of ***F***_h_ and ***v***_h_ are ***F***_h_:[*F*_min_ *F*_max_] and *v*_h_:[*v*_min_ *v*_max_], respectively. Considering the directionality of force and velocity, ***F***_h_ can be divided into seven fuzzy subsets {NB (Negative Big), NM (Negative Medium), NS (Negative Small), ZO (Zero), PS (Positive Small), PM (Positive Medium), and PB (Positive Big)}, and the end-effector velocity ***v***_h_ can be divided into seven fuzzy subsets {NB (Negative Big), NM (Negative Medium), NS (Negative Small), ZO (Zero), PS (Positive Small), PM (Positive Middle), PB (Positive Big)}. Similarly, the virtual damping ***C***_v_ can be divided into five fuzzy subsets: {VS (Very Small), S (Small), M (Medium), B (Big), VB (Very Big)}.

The triangular membership function is employed to fuzzify ***F***_h_, ***v***_h_, and ***C***_v_. The parameters in the membership function are obtained experimentally, and the distances between the peaks of ***F***_h_ and ***v***_h_ are, respectively, 3 and 0.04. The designed membership functions of ***F***_h_, ***v***_h_, and ***C***_v_ are depicted in Figure 3 and Figure 4. A fuzzy rule table is then developed, as shown in Table 1.

The virtual damping is obtained through defuzzification by using the discrete centroid method:(8)$$C_{v}=\frac{\sum \left(\mu_{C}\left(c\right)c_{pk}\right)}{\sum \mu_{C}\left(c\right)}.$$
where *c_pk_* denotes the horizontal coordinates of the peak of the membership function with virtual damping, *k* = 1, 2, 3, 4, 5.

By using (8), the human motion intention can be predicted in terms of the interaction force and end-effector velocity, and then the effective virtual damping can be obtained using (7). Therefore, variable admittance control is realized by adjusting ***C***_v_ based on the fuzzy theory and ***M***_v_ by using (6) to obtain the designed end-effector velocity with good compliance performance.

### 3.2. Velocity Feedforward Compensation Based on the State Observer

In the human–robot interaction process, the manipulator reacts according to the interaction force, and the robot’s inertia inevitably offers resistance to the human hand, resulting in delayed responses. Therefore, a state observer is designed to predict the joint torque due to the interaction force by considering the current motion state, and a velocity feedforward controller is proposed to append a compensatory velocity to the end-effector and improve system compliance.

During assisted human walking with a mobile manipulator, the desired velocity induced by the interaction force from the human hand is allocated to the manipulator and the mobile platform, respectively. Compliance is mainly reflected in the interaction between the manipulator and the human user. Thus, the effect of the mobile platform on the compliance of the interaction system can be neglected.

Based on the above assumptions, a state observer is designed for the manipulator to predict the joint torque induced by the interaction force. The dynamic model of the manipulator can be written as follows:(9)$$\left\{\begin{array}{c} M\left(q_{m}\right){\ddot{q}}_{m}+N\left(q_{m},{\dot{q}}_{m}\right)=\tau_{m}+\tau_{ext}\\ N\left(q_{m},{\dot{q}}_{m}\right)=C\left(q_{m},{\dot{q}}_{m}\right){\dot{q}}_{m}+G\left(q_{m}\right)+F_{f}\left(q_{m},{\dot{q}}_{m}\right) \end{array}\right..$$
where $M\left(q_{m}\right)\in R^{m\times m}$ is the inertia term of the manipulator, $C\left(q_{m},{\dot{q}}_{m}\right)\in R^{m\times m}$ is the centrifugal and Coriolis term, $G\left(q_{m}\right)\in R^{m\times 1}$ is the gravity term, $F_{f}\left(q_{m},{\dot{q}}_{m}\right)\in R^{m\times 1}$ is the friction term, $\tau_{m}\in R^{m\times 1}$ is the joint driving torque of the manipulator, and $\tau_{ext}\in R^{m\times 1}$ is the joint torque generated due to the external force acting on the end-effector.

According to (9), when the manipulator is subjected to the interaction force exerted by the human user, the joint torque caused by the external force can be expressed as follows:(10)$$\tau_{ext}=M\left(q_{m}\right){\ddot{q}}_{m}+N\left(q_{m},{\dot{q}}_{m}\right)-\tau_{m}.$$

Based on (10), a nonlinear disturbance state observer can be constructed to estimate the joint torque generated by the external force. The estimated values of ***M***(***q****_m_*) and $N\left(q_{m},{\dot{q}}_{m}\right)$ are $\hat{M}\left(q_{m}\right)$ and $\hat{N}\left(q_{m},{\dot{q}}_{m}\right)$, respectively, and their uncertainty factors are $\tilde{M}$, $\tilde{N}$, and $\hat{M}\left(q_{m}\right)$ is a symmetric positive-definite matrix as well. We can obtain
(11)$$\left\{\begin{array}{c} M\left(q_{m}\right)=\hat{M}\left(q_{m}\right)+\tilde{M}\\ N\left(q_{m},{\dot{q}}_{m}\right)=\hat{N}\left(q_{m},{\dot{q}}_{m}\right)+\tilde{N} \end{array}\right..$$

A disturbance vector $\tau_{d}\in R^{m\times 1}$ is designed as follows:(12)$$\tau_{d}=\tau_{ext}-\tilde{M}\ddot{q}-\tilde{N}.$$

If there are adequate and reasonable estimates of ***M***(***q***_m_) and $N\left(q_{m},{\dot{q}}_{m}\right)$, the joint torque ***τ****_ext_* of the manipulator due to the external force is almost equal to the disturbance vector ***τ****_d_*. By combining (10)–(12), we can obtain:(13)$$\tau_{d}=\hat{M}\left(q_{m}\right){\ddot{q}}_{m}+\hat{N}\left(q_{m},{\dot{q}}_{m}\right)-\tau_{m}.$$

Thus, the following state observer is proposed to observe the joint torque ***τ****_ext_*.
(14)$${\dot{\hat{\tau }}}_{d}=-L{\hat{\tau }}_{d}+L\left(\hat{M}\left(q_{m}\right){\ddot{q}}_{m}+\hat{N}\left(q_{m},{\dot{q}}_{m}\right)-\tau_{m}\right).$$
where ***L*** is the observer gain matrix.

In practice, to obtain an accurate ${\ddot{q}}_{m}$, the first-order difference of ${\dot{q}}_{m}$ and the Kalman filter are employed here.

Then, to verify the stability of the proposed state observer, its error vector $e_{dis}\in R^{m\times 1}$ is defined as follows:(15)$$e_{dis}=\tau_{d}-{\hat{\tau }}_{d}.$$

The change rate of this error is as follows:(16)$${\dot{e}}_{dis}={\dot{\tau }}_{d}-{\dot{\hat{\tau }}}_{d}.$$

During assisted walking, the mobile manipulator moves at a low velocity, and the disturbance vector varies slowly. In [37], the designed observer in (14) can be regarded as a non-dynamic nonlinear disturbance observer (NDOB). Therefore, it is reasonably assumed that:(17)$${\dot{\tau }}_{d}=0.$$

From (14), (16), and (17), we can obtain that:(18)$${\dot{e}}_{dis}={\dot{\tau }}_{d}-{\dot{\hat{\tau }}}_{d}=L{\hat{\tau }}_{d}-L\left(\hat{M}\left(q_{m}\right){\ddot{q}}_{m}+\hat{N}\left(q_{m},{\dot{q}}_{m}\right)-\tau_{m}\right)=L{\hat{\tau }}_{d}-L\tau_{d}.$$

From (15) and (18), we can obtain the following expression:(19)$${\dot{e}}_{dis}+Le_{dis}=0.$$

By ensuring that the proposed observer is asymptotically stable over a broad range, we define ***L*** as follows:(20)$$L=\mathrm{diag}{\left\{\begin{array}{cccc} \nu & \nu & \cdots & \nu \end{array}\right\}}_{m\times m}.$$
where ν is a positive constant.

Therefore, the proposed state observer (14), constrained by (20), is stable and effective. Then, we can obtain a reasonable ***τ****_ext_* by employing the observer (14).

Based on (14), the following feedforward velocity is designed to eliminate the robot’s inertia.
(21)$${\dot{q}}_{f}=TK_{f}\left({\hat{\tau }}_{d},{\dot{\hat{\tau }}}_{d}\right){\hat{M}}^{-1}\left(q_{m}\right){\hat{\tau }}_{d}.$$
where $K_{f}\left({\hat{\tau }}_{d},{\dot{\hat{\tau }}}_{d}\right)$ is the feedforward gain matrix, and T is the control period.

From (21), we can obtain ${\dot{q}}_{f}$ with the help of ${\hat{\tau }}_{d},{\dot{\hat{\tau }}}_{d}$, which can be used to improve the compliance of interactive systems.

However, upon the direct use of (21), $\tilde{M}$ and $\tilde{N}$ induced notable manipulator jitter in the experiments. Because ${\ddot{q}}_{m}$ is obtained after Kalman filtering, $\tilde{M}$ has little effect on this phenomenon, and the $\tilde{N}$ is the main driver of this phenomenon, possibly because of the friction torque and joint vibration. In addition, the position and velocity of the end-effector are mainly determined by the first three joints of the manipulator, and the latter three joints mainly affect the posture of the end-effector. Additionally, these noises have stronger effects on the latter three joints of the manipulator because of their small inertia. Therefore, to avoid unreasonable movements of the latter three joints and prevent manipulator jitter, the feedforward velocity compensation is applied only to the first three joints of the manipulator in this study. Herein, we can define $K_{f}\left({\hat{\tau }}_{d},{\dot{\hat{\tau }}}_{d}\right)$ in (21) as
(22)$$\left\{\begin{array}{c} K_{f}\left({\hat{\tau }}_{d},{\dot{\hat{\tau }}}_{d}\right)=\mathrm{diag}\left\{\begin{array}{llllll} k_{f1} & k_{f2} & k_{f3} & 0 & 0 & 0 \end{array}\right\}\\ k_{fj}=\left\{\begin{array}{ll} 1 & \begin{array}{ccc} {\dot{\hat{\tau }}}_{du}\leq \left|{\dot{\hat{\tau }}}_{dj}\right| & \mathrm{or} & {\hat{\tau }}_{dl}\leq \left|{\hat{\tau }}_{dj}\right| \end{array}\\ \frac{1}{2}\left(\sin\left(s_{a}\left|{\dot{\hat{\tau }}}_{dj}\right|+s_{b}\right)+1\right) & {\dot{\hat{\tau }}}_{dl}<\left|{\dot{\hat{\tau }}}_{dj}\right|<{\dot{\hat{\tau }}}_{du},\left|{\hat{\tau }}_{dj}\right|<{\hat{\tau }}_{dl}\\ 0 & \left|{\dot{\hat{\tau }}}_{dj}\right|<{\dot{\hat{\tau }}}_{dl},\left|{\hat{\tau }}_{dj}\right|<{\hat{\tau }}_{dl} \end{array}\right.,\;j=1,2,3 \end{array}\right..$$
where ${\hat{\tau }}_{dl}$ is the joint torque threshold to activate the process of velocity feedforward; ${\dot{\hat{\tau }}}_{dl},\;{\dot{\hat{\tau }}}_{du}$ are the thresholds of the change rate for the joint torques to initiate the process of velocity feedforward; *s*_a_ and *s*_b_ are smoothing factors; and *j* denotes the *j*-th joint of the manipulator.

According to (22), the process of velocity feedforward is activated only when ${\hat{\tau }}_{d}$ is relatively large or changes rapidly. Thus, the adverse effects of uncertain factors on manipulator motion can be avoided.

According to (3) and (21), ${\dot{q}}_{f}$ is mapped to $\Delta v_{f}$ in the Cartesian space as follows:(23)$$\Delta v_{f}=J_{m}\left(q_{m}\right){\dot{q}}_{f}$$

To guarantee that the mobile manipulator can better assist human walking, the end-effector of the manipulator should track the human user’s motion, such that the interactive system simultaneously has good compliance and motion-tracking performance. Moreover, it should also provide a certain vertical force to support the human hand while walking. Therefore, the magnitudes of ***v***_d_ in the x and y-directions are computed using the proposed variable admittance controller (4), while the velocity in the z-direction is computed using the following constant admittance controller:(24)$$M_{z}\ddot{z}+B_{z}\dot{z}+K_{z}\left(z-z_{d}\right)=F_{z}.$$

Finally, the proposed variable admittance controller can be expressed as follows:(25)$$\left\{\begin{array}{l} \begin{array}{l} M_{x}\ddot{x}+C_{x}\dot{x}=F_{x}\;,\;\mathrm{in}\;x\;\mathrm{direction}\\ M_{y}\ddot{y}+C_{y}\dot{y}=F_{y}\;,\;\mathrm{in}\;y\;\mathrm{direction} \end{array}\\ M_{z}\ddot{z}+B_{z}\dot{z}+K_{z}\left(z-z_{d}\right)=F_{z}\;,\;\mathrm{in}\;z\;\mathrm{direction} \end{array}\right..$$

The desired end-effector velocity required for the mobile manipulator is then modified as $v_{r}=v_{d}+\Delta v_{f}$.

Then, the desired ***v***_r_ can be expressed as follows:(26)$$v_{r}=\left[\begin{array}{c} v_{x}\\ v_{y}\\ v_{z} \end{array}\right]=\left[\begin{array}{c} \dot{x}\\ \dot{y}\\ \dot{z} \end{array}\right].$$

Therefore, *M*_x_ and *M*_y_ and *C*_x_ and *C*_y_ in (25) are revised based on (6)–(8). Then, ***v***_r_ is projected onto ***u***_b_ and ***u***_m_ by (3). In this manner, compliant interaction between the manipulator and the human user is realized, and the mobile manipulator can move according to the human motion intention.

## 4. Configuration Optimization of Mobile Manipulator

Given that the total DOF of the mobile manipulator when considering the nonholonomic constraints is b + r = 8, its Jacobian matrix ***J***_u_(6 × 8) is non-square. Because of the existing redundant DOFs of the mobile manipulator, there may exist infinite sets of joint velocities for a desired end-effector velocity in Cartesian space. 

To guarantee that the mobile manipulator has the optimal configuration when tracking the desired end-effector commands (***v***_r_), it is necessary to implement de-redundancy to realize motion control of the mobile manipulator. 

In this study, a null-space controller based on the projected gradient method is proposed.

### 4.1. Null-Space Controller

Considering ***v***_r_ as the desired command for the mobile manipulator, the secondary task is to implement an optimal configuration that can be defined on the basis of the null space of the mobile manipulator. The redundant solution of the mobile manipulator in (3) can be expressed as follows:(27)$$u_{rn}=J_{u}^{†}v_{r}+\left(I-J_{u}^{†}J_{u}\right)u_{n}.$$
where $J_{u}^{†}$ is the pseudo-inverse of the Jacobian matrix of the mobile manipulator, $I-J_{u}^{†}J_{u}$ is the orthographic projection operator representing the null space 𝒩$\left(J_{u}\right)$ of the mobile manipulator, and ***u***_n_ is the joint velocity vector required to perform secondary tasks.

The goal of the null-space control herein is to ensure that the mobile manipulator maintains its optimal configuration. Therefore, (27) is directly used to optimize the configuration of the mobile manipulator under null-space control. 

(1)Before the manipulator reaches the optimal configuration, the desired velocity ***u****_rn_* is converted into the commands ***u****_b_* and ***u****_m_* for the mobile robot and the manipulator through *J_u_*(*q*)^−1^. The mobile robot and the manipulator move, respectively, with the decomposed commands decided by (3) and (27).(2)After reaching the optimal configuration, ***u***_m_ along the horizontal direction is mostly distributed to the moving platform; the velocity in the vertical direction should be close to zero to ensure the provision of vertical support forces.

### 4.2. Null-Space Tasks

In this section, the projected gradient method is employed to solve the null-space tasks in (27) by designing an optimization function *F*(*u*). Then, with $u_{n}=\nabla_{u}F\left(u\right)$, we can implement the required secondary tasks. 

As a secondary task, ***u***_n_ is constantly projected onto the null space 𝒩$\left(J_{u}\right)$ of ***J****_u_*, and it does not affect the designed task ***v***_r_ in the task space. Here, *F*(***u***) should be a differentiable function.

To design a feasible *F*(***u***), we mainly consider the following three aspects:(1)The singular configuration cannot appear when the mobile manipulator flexibly interacts with the human user, meaning that the manipulator should be as far away from the singularity as possible.(2)The manipulator can adequately withstand external forces from the human user’s hand.(3)The effect of the manipulator’s gravity on itself should be as small as possible to ensure that the manipulator can adequately support the human hand in the z-direction.

For the first target, the joint velocity of the manipulator can be considered as the following velocity ellipsoid [38]:(28)$$E_{1}\left(q_{m}\right)=\sqrt{\left|J_{m}J_{m}^{T}\right|}.$$

The velocity ellipsoid can be the operational evaluation index of the manipulator. The larger the value of *E*_1_, the wider the non-singular configuration range of the manipulator. For convenient calculation, the square of the velocity ellipsoid is taken as the first objective function:(29)$$F_{1}\left(q_{m}\right)=\left|J_{m}J_{m}^{T}\right|.$$

For the second target, the external force can be considered from the manipulator force ellipsoid [38]. However, the force ellipsoid and the velocity ellipsoid are reciprocals of each other [39]. If the capacity of the end-effector to withstand external forces is increased, the corresponding improvement in the force ellipsoid will cause the manipulator to move closer to the singularity configuration. Therefore, it would be impossible to improve the capacity of the end-effector to withstand external forces as the manipulator moves farther away from the singular configuration. Thus, the force ellipsoid and the velocity ellipsoid should be balanced.

To ensure that the mobile manipulator can assist human walking, the interaction force between the end-effector and the human hand should be as small as possible in the horizontal direction, and it should perform a supporting role in the vertical direction. Therefore, we only need to increase the force capacity in the vertical direction.

The normalized vector of the manipulator joint torque is defined as follows:(30)$$\left\{\begin{array}{c} \tau_{\mathrm{norm}}=W_{\lim}\tau_{m}\\ W_{\lim}=\mathrm{diag}\left\{\begin{array}{cccc} \frac{1}{\tau_{m\lim1}} & \frac{1}{\tau_{m\lim2}} & \cdots & \frac{1}{\tau_{m\lim m}} \end{array}\right\} \end{array}\right..$$
where ***τ***_norm_ is the normalized joint torque vector, ***W***_lim_ is the proportional matrix considering the joint torque limits, and ***τ***_mlim*j*_ is the limit torque of joint j.

With the definition of force ellipsoid in [38], the constraint for the manipulator joint torques is defined as follows:(31)$${\left\|\tau_{\mathrm{norm}}\right\|}_{2}^{2}=\tau_{\mathrm{norm}}^{T}\tau_{\mathrm{norm}}\leq 1.$$

When the end-effector of the manipulator is subjected to an external force, ***F***_ext_, the joint torque is mainly induced by ***F***_ext_ and the manipulator’s own gravity, as follows:(32)$$\tau_{\mathrm{norm}}=J_{m}^{T}F_{ext}+G\left(q_{m}\right).$$

Then, we can project the gravity component ***G***(***q***_m_) in (32) onto its Cartesian space.
(33)$$\left\{\begin{array}{c} \tau_{\mathrm{norm}}=J_{m}^{T}\left(F_{ext}+G_{m}\right)\\ G\left(q_{m}\right)=J_{m}^{T}G_{m} \end{array}\right..$$

By substituting (33) in (31), we have
(34)$$\tau_{\mathrm{norm}}^{T}\tau_{\mathrm{norm}}={\left(F_{ext}+G_{m}\right)}^{T}J_{m}W_{\lim}W_{\lim}J_{m}^{T}\left(F_{ext}+G_{m}\right).$$

According to (34), if $\tau_{\mathrm{norm}}^{T}\tau_{\mathrm{norm}}$ is constant, meaning that the manipulator joint torque is constant, a reduction in $J_{m}W_{\lim}W_{\lim}J_{m}^{T}$ can lead to an increase in the external force that the manipulator can bear. However, directly considering $J_{m}W_{\lim}W_{\lim}J_{m}^{T}$ reduction as the optimization goal will increase the force capacity of the manipulator in all directions [40], which may weaken the force application ability in the vertical direction under the constraints of joint maximum torques. Moreover, there are different requirements for force capacity in the horizontal and vertical directions during assisted walking. 

Therefore, it is assumed that the end-effector has the equivalent stiffness $K_{opt}\in R^{3\times 3}$, and the stiffness *K_op_*_t_ of the end-effector is obtained through the pretests to guarantee it can supply enough vertical force under a permissible vertical deformation. The virtual deformation of the end-effector due to ***F****_ext_* and ***G****_m_* are $\Delta x_{ext}=F_{ext}/K_{opt}$ and $\Delta x_{g}=G_{m}/K_{opt}$, respectively. Then, by substituting these expressions into (34), we obtain
(35)$$\tau_{\mathrm{norm}}^{T}\tau_{\mathrm{norm}}={\left(\Delta x_{ext}+\Delta x_{g}\right)}^{T}K_{opt}J_{m}W_{\lim}W_{\lim}J_{m}^{T}K_{opt}\left(\Delta x_{ext}+\Delta x_{g}\right).$$

According to (35), by reducing $K_{opt}J_{m}W_{\lim}W_{\lim}J_{m}^{T}K_{opt}$, we can increase the maximum deformation of the end-effector by considering the external forces along different directions, thereby increasing the capacity of the end-effector to withstand external forces. Then, the optimization function of the second objective along the vertical direction that is needed to increase the capacity of the end-effector to withstand external forces can be expressed as follows:(36)$$F_{2}\left(q_{m}\right)={\left(\delta \left(K_{opt}J_{m}W_{\lim}W_{\lim}J_{m}^{T}K_{opt}\right)\delta \right)}^{-1}.$$
where $\delta \in R^{3\times 1}$ is the specified optimized direction vector for the enhanced force capacity, and ***δ*** = [0, 0, 1]^T^.

According to (35), the self-gravity of the end-effector affects its capacity to withstand external forces in the vertical direction. Therefore, to optimize the gravity-induced deformation, it is reasonable to set the third objective function as follows:(37)$$F_{3}\left(q_{m}\right)=\Delta x_{g}^{T}\Delta x_{g}.$$

Thus, we can decrease the effect of ***G***(***q***_m_) in (32) by reducing the objective function *F*_3_ in (37), meaning that we can reduce the joint torques induced by ***G***(***q***_m_) by reducing $\Delta x_{g}$.

During optimization, the three functions in (29), (36), and (37) should have the same order of magnitude; then, the overall optimization function can be expressed as follows:(38)$$\left\{\begin{array}{c} \max F\left(q_{m}\right)=w_{1}\alpha_{1}F_{1}+w_{2}\alpha_{2}F_{2}-w_{3}\alpha_{3}F_{3}\\ \begin{array}{r} s.t.\tau_{j}\in \left[\tau_{j\min},\tau_{j\max}\right]\\ q_{j}\in \left[q_{j\min},q_{j\max}\right] \end{array} \end{array}\right..$$
where *w*_1_, *w*_2_, and *w*_3_ are the weights of the three optimization functions; *α*_1_, *α*_2_, and *α*_3_ are the normalized factors; *τ_j_*_min_, *τ_j_*_max_ are the minimum and maximum torque of the *j*-th joint; and *q_j_*_min_ and *q_j_*_max_ are the minimum and maximum angle of the *j*-th joint.

The velocity tasks defined by the null space can be expressed as follows:(39)$$u_{n}=\lambda \left[\begin{array}{c} 0_{b\times 1}\\ \nabla_{q_{m}}F \end{array}\right].$$
where *λ* is the control gain and $\nabla_{q_{m}}F$ is the gradient of *F*(***q***_m_) in (38).

By substituting (39) in (27), we obtain the following null-space controller:(40)$$u_{rn}=J_{u}^{†}\times \left[\begin{array}{c} \dot{x}\\ \dot{y}\\ \dot{z}\\ 0_{3\times 1} \end{array}\right]+\left(I-J_{u}^{†}J_{u}\right)\times \left[\begin{array}{c} 0_{b\times 1}\\ \lambda \times \nabla_{q_{m}}F \end{array}\right].$$

### 4.3. Position and Posture Controller for the End-Effector

During assisted walking, the position and posture of the end-effector should be maintained at their optimal configuration. Therefore, we apply a proportional controller to the end-effector. 

The quaternion in ideal position and posture of the end-effector ***Q***_de_ can be obtained from Rew in Tew. The desired end-effector position in Σ_b_ is set as ***x***_e_. Assuming that the current end-effector position is ***x***_c_, the error between ***x***_e_ and ***x***_c_ is ***x***_err_ = ***x***_e_ − ***x***_c_. Therefore, the desired linear velocity ***v***_de_ of the end-effector can be expressed as follows:(41)$$v_{de}=K_{v}x_{err}.$$
where *K*_v_ is the proportional coefficient.

The current posture of the end-effector can be expressed as ***Q***_c_ = *w*_c_ + ***u***_c_, and the error between ***Q***_de_ and ***Q***_c_ can be written as:(42)$$Q_{err}=Q_{de}Q_{c}^{*}.$$
where $Q_{c}^{*}=w_{c}-u_{c}$ is the conjugate of ***Q***_c_.

Then, the angle errors can be obtained from ***Q***_err_, as follows:(43)$$\theta_{err}=\left[\begin{array}{c} arctan(\frac{2(w_{err}x_{err}+y_{err}z_{err})}{1-2(x_{err}^{2}+y_{err}^{2})})\\ arcsin(2(w_{err}y_{err}-x_{err}z_{err}))\\ arctan(\frac{2(w_{err}z_{err}+x_{err}y_{err})}{1-2(y_{err}^{2}+z_{err}^{2})}) \end{array}\right].$$

Therefore, the desired angular velocity of the end-effector can be obtained.
(44)$$w_{de}=K_{w}\theta_{err}.$$
where *K*_w_ is the proportional coefficient.

The controller for the end-effector can be summarized as
(45)$$v_{e}=\left[\begin{array}{c} v_{de}\\ w_{de} \end{array}\right].$$

Then, ***v***_e_ is projected onto the mobile platform and the manipulator, separately, by using ***J***_u_(***q***) in (3):(46)$$u_{e}=\left[\begin{array}{c} u_{be}\\ u_{me} \end{array}\right]=J_{u}^{†}\times v_{e}.$$
where ***u***_be_ and ***u***_me_ are the velocity of the mobile platform and manipulator.

### 4.4. Integrated Controller for Assisted Walking

From (40) and (46), the velocity relationship between the mobile platform and the manipulator can be inferred clearly.

(1)In the optimization process, there is no interaction between the human user and the mobile manipulator during optimization. Therefore, the velocities of the mobile platform and manipulator can be computed as follows:


(47)
$$\left[\begin{array}{c} u_{b}\\ u_{m} \end{array}\right]=\left(I-J_{u}^{†}J_{u}\right)\times \left[\begin{array}{c} 0_{b\times 1}\\ \lambda \times \nabla_{q_{m}}F \end{array}\right]+J_{u}^{†}\times \left[\begin{array}{c} v_{de}\\ w_{de} \end{array}\right].$$


(2)During assisted walking, the velocities of the mobile platform and manipulator can be obtained using the total velocity controller expressed as (47):


(48)
$$\left[\begin{array}{c} u_{b}\\ u_{m} \end{array}\right]=J_{u}^{†}\times \left[\begin{array}{c} \dot{x}\\ \dot{y}\\ \dot{z}\\ 0_{3\times 1} \end{array}\right]+\left(I-J_{u}^{†}J_{u}\right)\times \left[\begin{array}{c} 0_{b\times 1}\\ \lambda \times \nabla_{q_{m}}F \end{array}\right]+J_{u}^{†}\times \left[\begin{array}{c} v_{de}\\ w_{de} \end{array}\right].$$


The assisted walking system comprising the mobile manipulator is developed, as depicted in Figure 5. In the following section of null-space optimization, the desired velocity of the end-effector can be obtained by (47), and the mobile platform and the manipulator move together. After null-space optimization, the end-effector maintains its posture by the proposed controller in (45), and therefore, the desired velocity of the end-effector in the horizontal direction is allocated to the mobile platform while that in the vertical direction is set to be 0 to provide adequate assisted walking support to the user. The mobile manipulator walking-assistance technology based on the proposed velocity feedforward compensator in (21) and variable admittance controller in (25) can be implemented. It is worth noting that the configuration optimization is carried out first by the objective function (38) and the null-space controller (40), and the human–robot interaction begins after the optimization is completed. Then, the optimized configuration can be maintained by (40) in the process of assisted walking.

## 5. Experimental Verification

In order to measure the compliance, stability, and comfort of the human–robot interaction, we will collect the velocity, the interaction force, the trajectory of the end-effector, and the joint angle of the manipulator in the interaction process shown in Appendix A. Due to the limitations of the laboratory conditions, we selected three volunteers to test the different requirements of walking assistance.

### 5.1. Experimental Setup

The experimental platform consists of a UR3e manipulator (Universal Robots Inc., Aarhus, Denmark) and a four-wheeled mobile robot named SCOUT (AgileX Inc., Dongguan, China), as shown in Figure 6, and the parameters are summarized in Table 2. Herein, *L*, *W,* and *H* are the length, width, and height of the SCOUT robot; *l*_0_, *w*_0_, and *h*_0_ are the installation positions of the manipulator; and *l*_1_ and *l*_2_ are the wheelbase and wheel-track. The proposed assisted walking system is implemented in ROS (Robot Operation System) Kinetic, while the mobile robot owns an odometer, the manipulator owns a six-axis force sensor at the end-effector, and the motion for its joints can be obtained in real time. The global controller for the mobile robot and the manipulator are integrated on an upper computer based on Figure 5, which communicates with the lower control units for the mobile robot and the manipulator through CAN (Controller Area Network) and LAN (Local Area Network), respectively. The calculation frequency of the whole system is 500 Hz. The six-axis force sensor shown in Figure 6 is employed to obtain the interacting forces for the assisted walking system shown in Figure 5. The experiments can be seen in Appendix A.

First, a human-machine flexible interaction experiment and the following null-space control experiment are conducted. The effectiveness of the proposed controllers in (38) and (40) is reflected by the optimization functions *F*_1_, *F*_2_, *F*_3_, and *F* and the joint torques of the manipulator before and after optimization. 

Second, a compliant interactive experiment is conducted to verify the validity of the proposed state observer (14), the velocity feedforward compensator (21), and the variable admittance controller (25) based on the fuzzy theory. The effectiveness of the above control algorithm is reflected by the real and predictive values of joint torques, the interaction forces, and the velocities of the end-effector and mobile platform in the process of compliant interaction. 

Finally, different volunteers conducted the experiments to verify the stability and robustness of the assisted walking technology based on the compliant interaction approach proposed herein. The effectiveness of the approach is reflected in the interaction forces reported by the various volunteers and the end-effector velocities recorded during assisted walking.

### 5.2. Human-Machine Flexible Interaction Experiment

In this paper, the proposed human–robot flexible interaction system is verified by experiments. The specific test is to let the operator hold the end-effector, and then the operator pulls the manipulator to carry out a series of complex actions. The traditional admittance control (Method I [41]), admittance control (Method II [42]) with fuzzy adjustment of virtual damping, admittance control (Method III [31]) with fuzzy adjustment of virtual damping, and adjustment of virtual mass are tested together with the flexible interactive control method (Method IV) proposed in this paper, and among them, Methods I, II, and III do not have feedforward velocity control. The parameters of the fuzzy inference system and virtual mass control law used in Methods II and III are consistent with those of Method IV. The experimental results are shown in Figure 7, Figure 8 and Figure 9 and Table 1.

As shown in Table 3, the interaction forces in Method IV are much smaller, both at maximum and average, than in the previous three methods. Although the average value of the interaction force in Method II is slightly higher than that in Method I, the velocity of the end-effector is also higher than that in Method I, and it is more obvious. In addition, the maximum interaction force in Method II is smaller than that in Method I, while the maximum speed of the end-effector is larger. It can be seen that the human motion intention prediction method proposed in this paper can improve the compliance of the human–robot interaction system.

The maximum value and average value of interaction force in Method III are smaller than those in Method II, and the end-effector movement speed is faster, which proves that maintaining the dynamic response characteristics of the system can further improve the compliance of the interactive system. In addition, although the maximum and average velocity of the end-effector in Method IV are 4.62% and 1.75% lower than that in Method III, the maximum and average interaction force are 5.83% and 11.34% lower than that in Method III, respectively. Method IV can make the manipulator end effector produce a speed similar to Method III with less force due to the use of the feedforward velocity control. In summary, the Method IV proposed in this paper has good compliance and comfort among the four methods.

### 5.3. Null-Space Control Experiment

To verify whether the proposed objective function (38) can find an optimized configuration for the mobile manipulator while the end-effector is maintained at a given point.

The initial position of the joints is *q*_m1_ = [0, −2π/3, −π/2, −5π/6, −π/2, 0]^T^, and the position of the end-effector in Σ_w_ is *x*_e1_ = [0.7197, −0.1310, 0.1708]^T^. During the optimization, the position and posture controller (45) for the end-effector is employed. After the optimization, the final position of the joints is *q*_m2_ = [0, −89π/200, −31π/40, −781π/1000, −π/2, 0]^T^, and the position of the end-effector is *x*_e2_ = [0.7055, −0.1310, 0.1708]^T^. The experimental results are shown in Figure 6, Figure 7 and Figure 8.

Figure 9 shows the configurations of the mobile manipulator before and after optimization. According to Figure 10, in the initial posture, the three objective functions are close. As the optimization proceeds, *F*_2_ in (36) and *F*_1_ in (29) exhibit two opposite trends, indicating that enhancement of the force capacity of the end-effector and maintenance of distance from the singular configuration cannot be achieved simultaneously. However, the *F*_3_ in (37) decreases rapidly and then increases marginally to reach a small steady-state value, indicating that the gravity effect of the manipulator is effectively suppressed, and the overall objective function *F* (38) gradually converges to a stable value.

The minimum value of *F*_1_ is greater than 0, indicating it can avoid singularity. The minimum value of *F*_3_ is smaller than its stable value, indicating that if the force capacity of the end-effector in the vertical direction is improved, the influence of gravity can be decreased, but it cannot be eliminated.

As shown in Figure 11, the pose of the end-effector may deviate marginally from the desired pose. According to Figure 11a, the end-effector has different degrees of deviation in three directions, of which the position error in the x-direction is the greatest, and its maximum value is 1.492 cm. The obvious deviation in the x-direction may have been caused by the mobile platform, which cannot implement the desired small velocity commands. The end posture errors shown in Figure 11b are within an extremely small range (approximately 2 × 10^−4^ rad). Therefore, it can be concluded that the mobile manipulator can well maintain the desired pose of the end-effector during the optimization, indicating that the null-space approach to finding an optimized configuration is effective.

To verify whether the proposed null-space controller (40) can improve the force capacity of the end-effector in the vertical direction while maintaining the end-effector pose, two comparison tests are conducted by applying the same load to the end-effector before and after optimization. An end gripper is used to grab a bottle of water (approximately 550 mL), meaning that a vertical load of approximately 6 N is applied to the end-effector, as shown in Figure 12.

As shown in Figure 13, before the optimization, the heavy load is grabbed by the manipulator gripper at approximately 35 s, and after the optimization, it is grabbed at approximately 27 s. Regardless of whether the end-effector is subjected to an external force, after optimization, the torques of joints 2 and 3 are significantly smaller than those before optimization, meaning that the proposed objective functions *F*_2_ (36) and *F*_3_ (37) are effective. Moreover, the torques of the other joints are always small because these joints are not used to balance the vertical load against gravity. The torque details of the joints are summarized in Table 4.

According to Table 4, when the manipulator grasps the load and remains static, the torques of joints 2, 3, and 4 vary substantially, and the 2-norm of the torque vector increases accordingly. The torques of joints 2, 3, and 4 are mainly used to balance the vertical load against gravity. In the absence of external load, the torques of joints 2, 3, and 4 decrease by 50.78%, 28.11%, and 55.02%, respectively, after optimization, meaning that the gravity effect of the manipulator itself is effectively weakened. Under a load, the torques of joints 2 and 3 are 63.07% and 14.18% smaller, respectively, than those before optimization, and the torque of joint 4 is close to that before optimization. In general, the 2-norm of the torque vector after optimization is 44.34% and 41.77% smaller than that before optimization, respectively. This indicates that the null-space controller (40) can significantly improve the force capacity of the end-effector in the vertical direction.

### 5.4. Assisted Walking Test

In the assisted walking test, the manipulator position is set to *q*_1_ = [0, −2π/3, −π/2, −π/3, −π/2, 0]^T^, and the end-effector position is *x*_1_ = [0.713, −0.131, 0.348]^T^. First, the null-space controller is used to ensure that the mobile manipulator reaches the optimal configuration. Then, the manipulator position is *q*_2_ = [0, −89π/200, −387π/500, −281π/1000, −π/2, 0]^T^, and the end-effector position is *x*_2_ = [0.698, −0.133, 0.348]^T^. Next, the experimenter’s arm is clamped using the gripper, and compliant force interaction is performed along the x-, y-, and z-directions of the mobile manipulator. The experimental process is shown in Figure 14.

Figure 15 shows the trajectory of the end-effector in the world coordinate system during assisted walking. The trajectory is smooth and consistent with the human motion intention. A complete circle is drawn in the horizontal plane, which verifies that the compliant interaction controller proposed in this paper can not only perceive and execute the human user’s forward or backward motion intention but also comply with the human user’s steering intention.

Figure 16 presents a comparison between the real (values observed using sensors) and predicted (values obtained using the state observer (14)) values of the joint torques induced by the human arm on the end-effector during assisted walking. Herein, the proposed velocity feedforward compensator (21) is only applied to the first three joints, and the root mean square errors of joints 1, 2, and 3 are 0.244, 0.171, and 0.154, respectively, and the mean square errors of the latter three joints are 0.236, 0.267, and 0.178, respectively. In conclusion, the errors between the predicted and real values are small for the first three joints, and none of the predicted values are greater than the corresponding real values, thereby guaranteeing that the manipulator moves within the allowable and safe range.

These findings validate the effectiveness of the proposed velocity feedforward controller (21) and the proposed state observer (14). Moreover, the predictions for joint torque are accurate and obtained in real time, and they can be used to further improve the performance of motion-tracking and the compliance between the human and the robot.

Figure 17 and Figure 18 present the human–robot interaction forces and end-effector velocity during assisted walking. The interaction forces in the x- and y-directions are less than 15 N, and the end-effector velocity can even exceed 0.3 m/s, meaning that the compliant interaction method proposed herein offers good compliance in the horizontal direction. At 65 s, a vertical load (of approximately 30 N) is applied to the end-effector, resulting in a small vertical velocity (about 0.02 m/s), which validates that the mobile manipulator equipped with the designed controller can provide reliable support to the human user’s hand.

As shown in Figure 19, when the end-effector is subjected to an interaction force in the x-direction (from 5 s to 22 s), the mobile platform tracks the desired velocity *u*_b_ in (48), which is forward or backward. The mobile platform then rotates to track the desired lateral motion or turning motion when the end-effector is subjected to an interaction force in the y-direction (from 22 s to 65 s). This is because the mobile platform used in this study is constrained by non-holonomic kinematics.

As shown in Figure 20, the joint angles of the manipulator vary in a small range because the proposed approach can maintain the optimal configuration by the controllers (40) and (45). This guarantees that the robot always has good force capacity and stability.

Figure 21 shows the position errors of the end-effector along the x, y, and z axes. The maximum position error along the x-direction (approximately 1.5 cm) is greater than that along the y-direction (approximately 0.2 cm) and z-direction (approximately 0.7 cm) from 5 s to 22 s. This is because the maximum interaction force along the x-direction (approximately 12 N) is greater than that along the y-direction (approximately 5 N), while a stiffness coefficient is applied along the z-direction in (25). From 22 s to 65 s, the proposed approach performs similarly in the y-direction. At 65 s, the end-effector is subjected to a large vertical load (approximately 30 N), and the displacement along the z-direction becomes approximately 1.7 cm. The displacement in the z-direction can generate an elastic force that can support the load in the vertical direction. However, the deviation varies within a small range (0–2 cm), which verifies the effectiveness of the proposed position and posture controller for the end-effector (45) in terms of controlling its position.

Figure 22 presents the posture errors of the end-effector relative to the mobile platform. The angle error around the z axis is approximately 0.04 rad, while others are about 0. This is because when the mobile manipulator is subjected to the interaction force in the y-direction, the mobile platform rotates around the z axis. Due to this force, an obvious error appears around the z-axis. However, the posture errors vary within a small range (0–0.04 rad), which verifies the validity of the proposed position and posture controller for the end-effector (45) in terms of controlling its posture.

### 5.5. Assisted Walking Tests with Three Volunteers

To further verify the adaptability of the assisted walking approach proposed herein, comparison tests are conducted. Herein, we select three volunteers who have different physical characteristics and can represent different groups of people. These volunteers are A (male), B (female), and C (male). Their heights and weights are 185 cm, 165 cm, and 175 cm, and 75 kg, 45 kg, and 60 kg, respectively. In the following experiments, the reference path is similar to a sine curve, as shown in Figure 23. This path is only a guideline for the volunteers, and it is not strictly tracked by humans. The experimental results are presented in Figure 24, Figure 25 and Figure 26, and the performances of the three volunteers are comparatively summarized in Table 5. Moreover, we interviewed three volunteers to evaluate the comfort, stability, and compliance of the assisted walking process, and the score range was 0–10 points, as shown in Table 6.

Through analyzing the experimental data of three different volunteers in Figure 24, Figure 25 and Figure 26, the velocity of the end-effector all presents the same change trend with the interaction force. In addition, by analyzing the data in Table 5, it can be seen that the absolute maximum force between three different volunteers and the mobile manipulator in the x-direction is 16.82 N, 15.48 N, and 13.51 N, respectively. The maximum absolute values of the end-effector velocity in the x-direction are 0.45 m/s, 0.37 m/s, and 0.32 m/s, respectively. It can be concluded that the bigger the interaction force is in a certain direction of the end-effector, the bigger the velocity of the end-effector is in that direction. It can be seen that the proposed controller (48) can produce a corresponding speed according to the interaction forces exerted by different users and obediently track the movement of the users. It indicates that the proposed assisted walking controller (48) is robust for users with different gaits.

In the vertical direction, the maximum absolute values of the interaction forces are 24.41 N, 27.32 N, and 28.69 N, respectively, and the corresponding maximum absolute values of the end-effector velocities in the z-direction are 0.0051 m/s, 0.0093 m/s, and 0.0096 m/s, respectively. Moreover, the velocities vary within a small range, as shown in Figure 24b, Figure 25b and Figure 26b. Therefore, during assisted walking, even if the end-effector bears a large vertical load, its deformation is small, meaning that the robot can provide adequate support force to the human user in the vertical direction. 

In Table 6, compared with the other two volunteers, volunteer C had the best experience during the assisted walking process and gave a high evaluation. Volunteer B, the only female, only had a high evaluation on stability. Volunteer A, who is the highest among the three volunteers, also had a good evaluation of the compliance, stability, and compliance of the assisted walking system. Based on their feedback, we can know that the assisted walking system has good compliance, stability, and robustness for different humans. However, they all mentioned that the comfort and compliance of the walking assistance system when turning is not as good as walking in a straight line, and we will further improve the comfort and compliance of assisted walking when turning in the future. 

The results of the three abovementioned experiments involving different volunteers indicate that the proposed assisted walking approach with the mobile manipulator can accurately identify human motion intentions and implement compliant tracking of human motion across individuals. In conclusion, the proposed approach has good stability, high accuracy, and a high level of robustness. Although the amplitudes of the interaction forces shown in Figure 24, Figure 25 and Figure 26 are small enough, this may not be suitable for a small number of people, and we will make further research on the amplitude control of the interactive forces.

## 6. Conclusions

In this paper, a novel approach to human–robot-compliant interaction for mobile manipulators is proposed. This approach can be used to assist human walking. To realize compliant interaction between a human user and the mobile manipulator, a method for predicting the human user’s motion intention based on the fuzzy theory is proposed, and it is used to regulate the mass and damping coefficients. Moreover, a velocity feedforward compensator with a state observer is used to reduce the resistance caused by the inertia of the mobile manipulator. Furthermore, a null-space controller based on the projection gradient method is proposed to maintain the optimal manipulator configuration during assisted walking. The experimental results obtained through experiments involving different volunteers indicated that the proposed approach has good stability, high accuracy, and a high level of robustness. 

At present, the aging society requires more assisted walking equipment for rehabilitation and daily nursing. As a multi-functional robot, the mobile manipulator can not only meet other daily services but also assist humans in walking, according to the proposed approach in this paper. In the future, we will seek cooperation from different kinds of people to extend the application of the proposed assisted walking system in different environments, including elderly people and hospital patients, and further research on the anti-fall safety strategies and improve its performance, which can make it safer and more intelligent and can be widely applied for elderly or disabled persons.

## Figures and Tables

**Figure 1 biomimetics-09-00104-f001:**
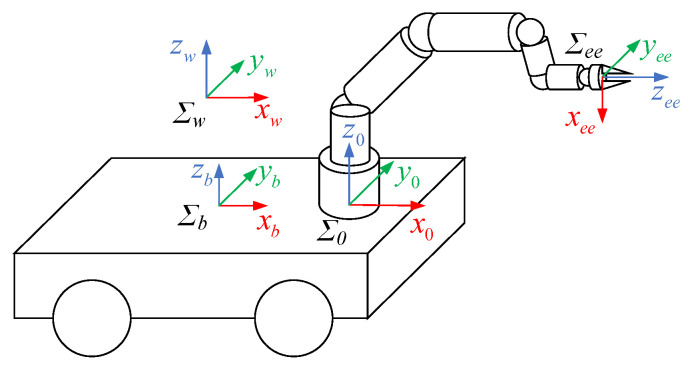
Schematic diagram of mobile manipulator system.

**Figure 2 biomimetics-09-00104-f002:**
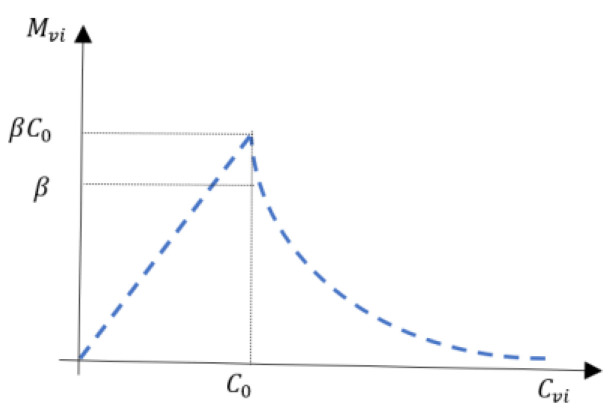
Schematic diagram of piecewise function in (6).

**Figure 3 biomimetics-09-00104-f003:**
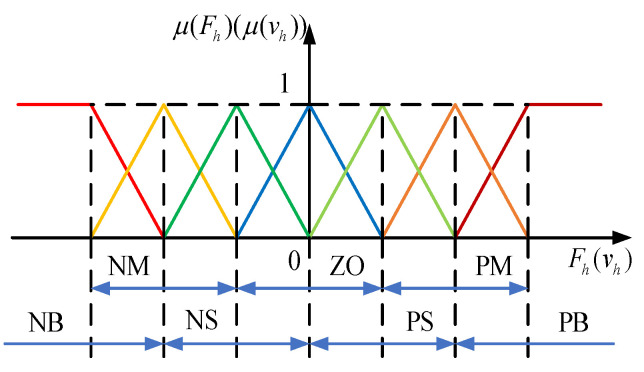
Membership function of interaction force and end-effector velocity.

**Figure 4 biomimetics-09-00104-f004:**
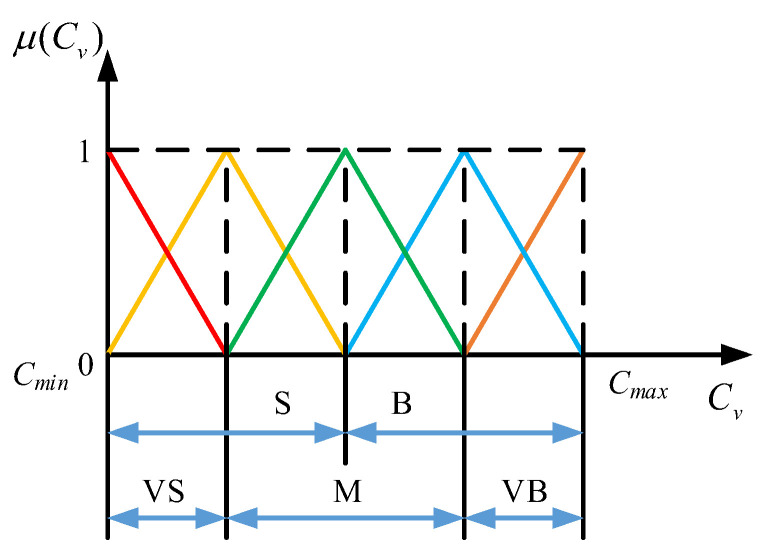
Membership function of virtual damping.

**Figure 5 biomimetics-09-00104-f005:**
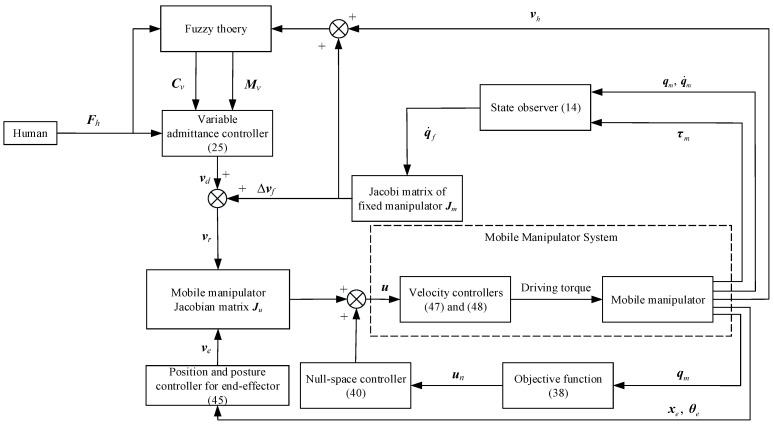
Assisted walking system with mobile manipulator.

**Figure 6 biomimetics-09-00104-f006:**
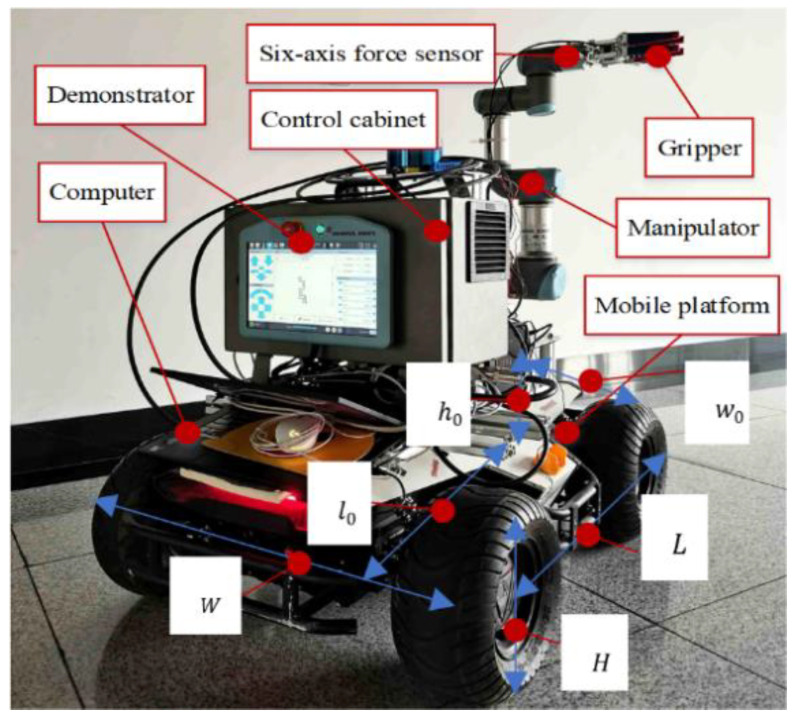
Mobile manipulator experimental system.

**Figure 7 biomimetics-09-00104-f007:**
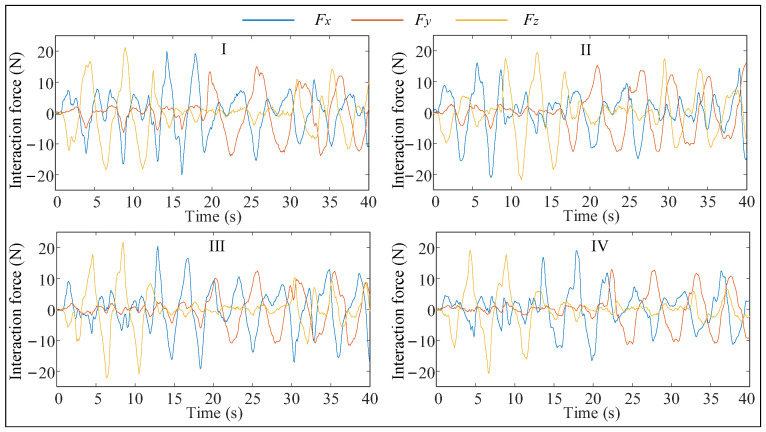
Interaction force changes during interaction.

**Figure 8 biomimetics-09-00104-f008:**
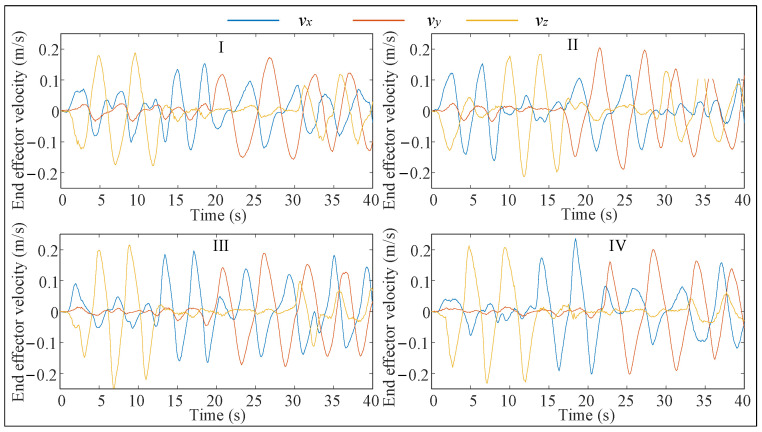
End effector velocity changes during interaction.

**Figure 9 biomimetics-09-00104-f009:**
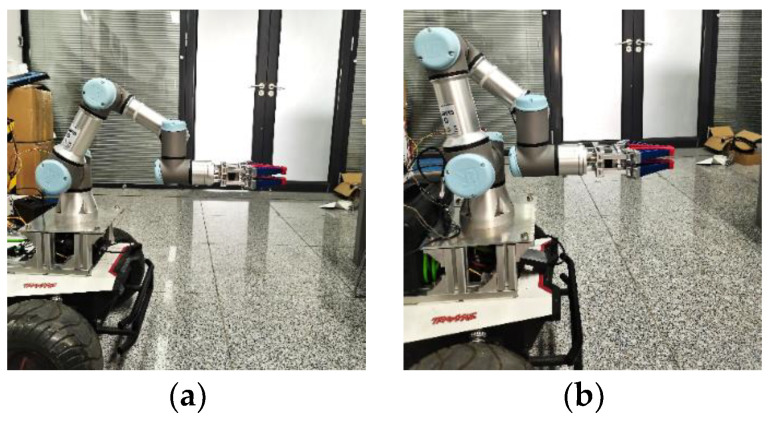
Comparison of robot configuration before and after optimization. (**a**) Before optimization; (**b**) After optimization.

**Figure 10 biomimetics-09-00104-f010:**
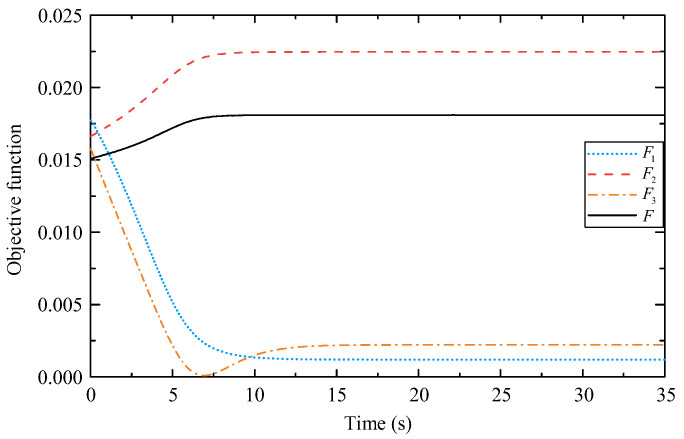
Changes in objective functions.

**Figure 11 biomimetics-09-00104-f011:**
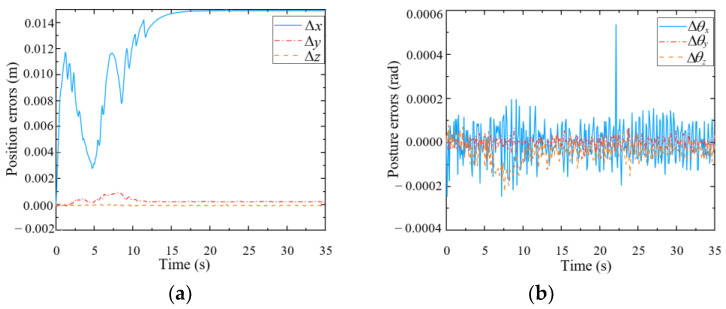
Pose errors of the end-effector during optimization. (**a**) Position errors; (**b**) Posture errors.

**Figure 12 biomimetics-09-00104-f012:**
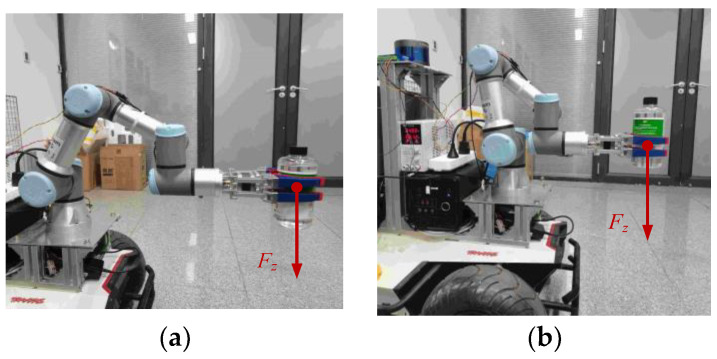
Comparison of force capacity before and after optimization. (**a**) Before optimization; (**b**) After optimization.

**Figure 13 biomimetics-09-00104-f013:**
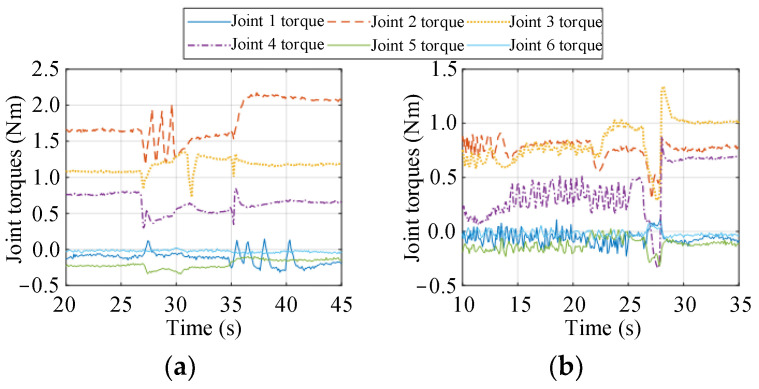
Joint torques before and after optimization. (**a**) Before optimization; (**b**) After optimization.

**Figure 14 biomimetics-09-00104-f014:**
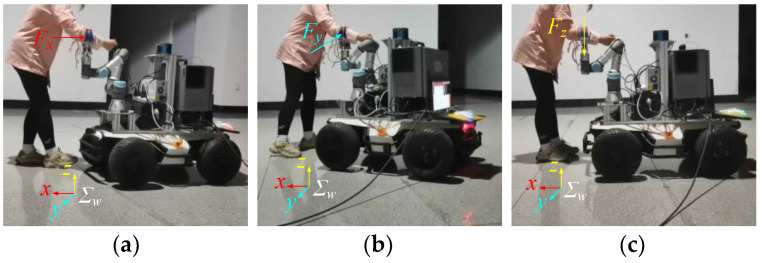
Human–robot compliant interaction tests. (**a**) x-direction; (**b**) y-direction; (**c**) z-direction.

**Figure 15 biomimetics-09-00104-f015:**
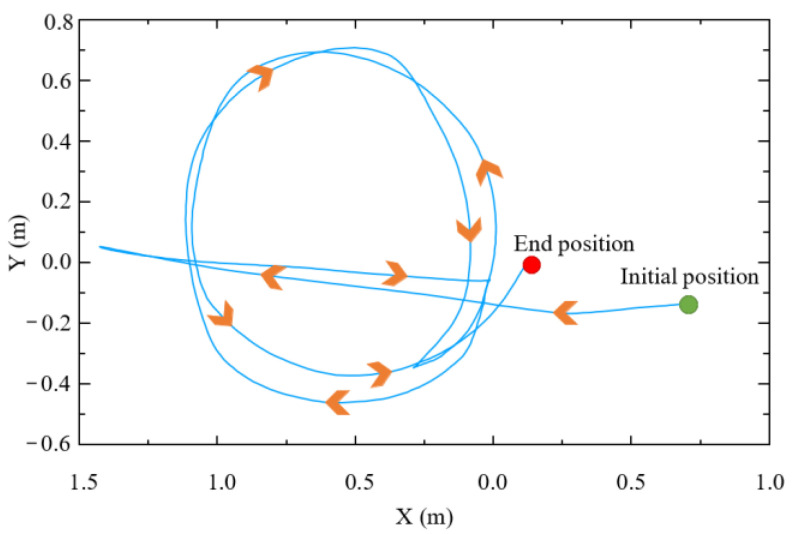
Trajectory of end-effector.

**Figure 16 biomimetics-09-00104-f016:**
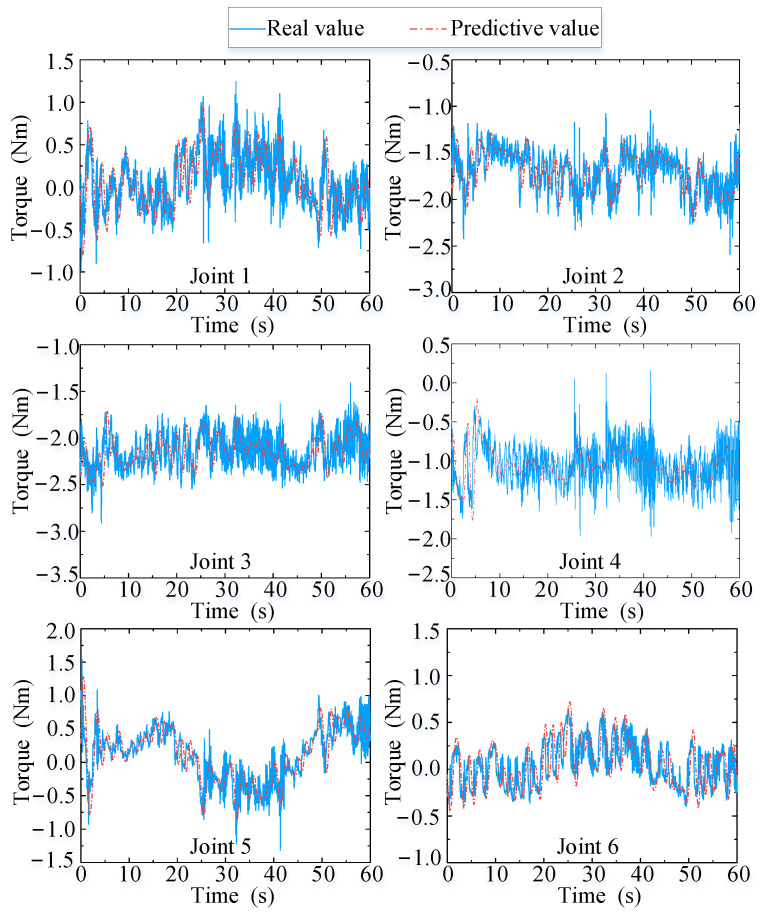
Real and predicted values of joint torque.

**Figure 17 biomimetics-09-00104-f017:**
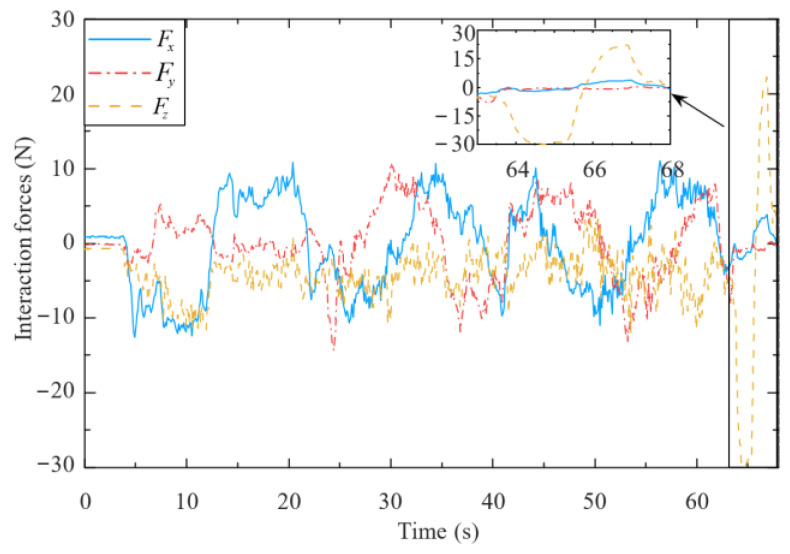
Interaction forces of end-effector (from 65 s, a vertical load (of approximately 30 N) is applied to the end-effector, resulting in a small vertical velocity (about 0.02 m/s in Figure 18)).

**Figure 18 biomimetics-09-00104-f018:**
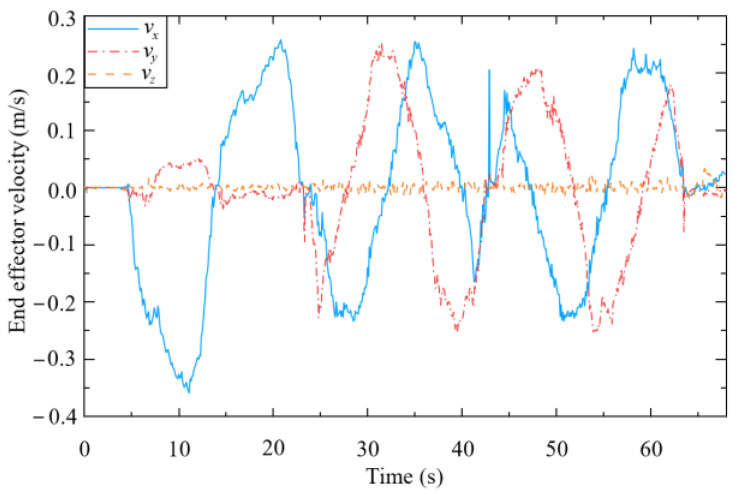
Velocity of end-effector.

**Figure 19 biomimetics-09-00104-f019:**
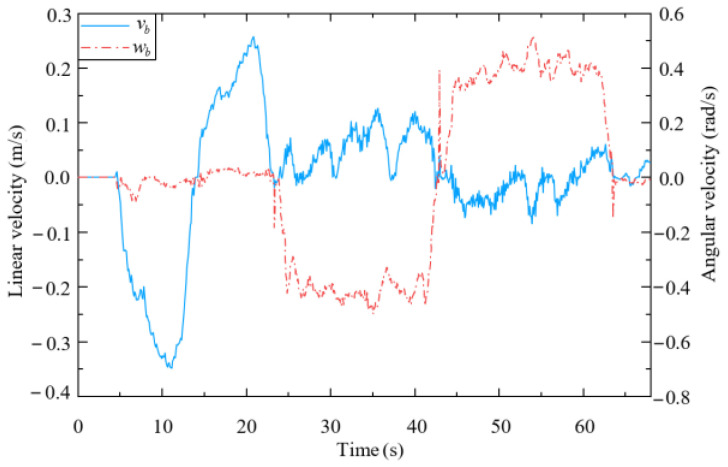
Velocity of mobile platform.

**Figure 20 biomimetics-09-00104-f020:**
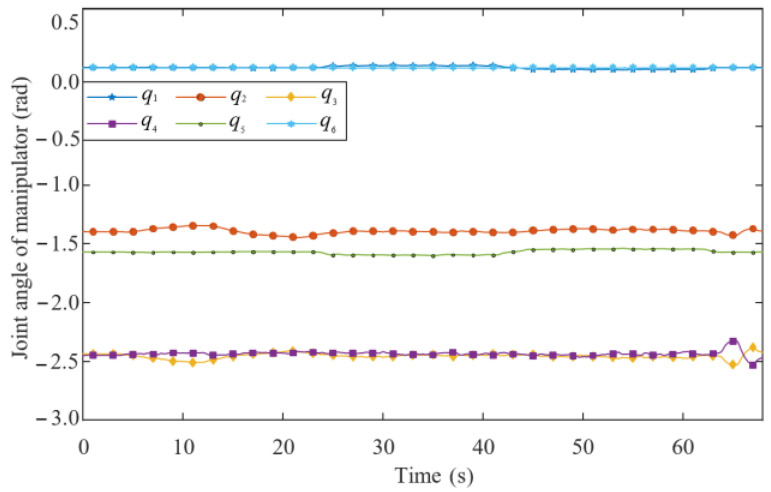
Angle of each joint.

**Figure 21 biomimetics-09-00104-f021:**
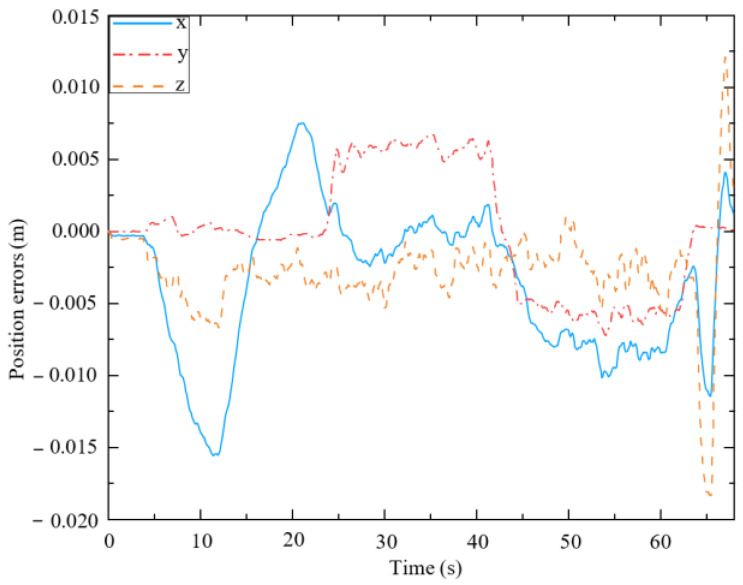
Position errors of the end-effector relative to the mobile platform.

**Figure 22 biomimetics-09-00104-f022:**
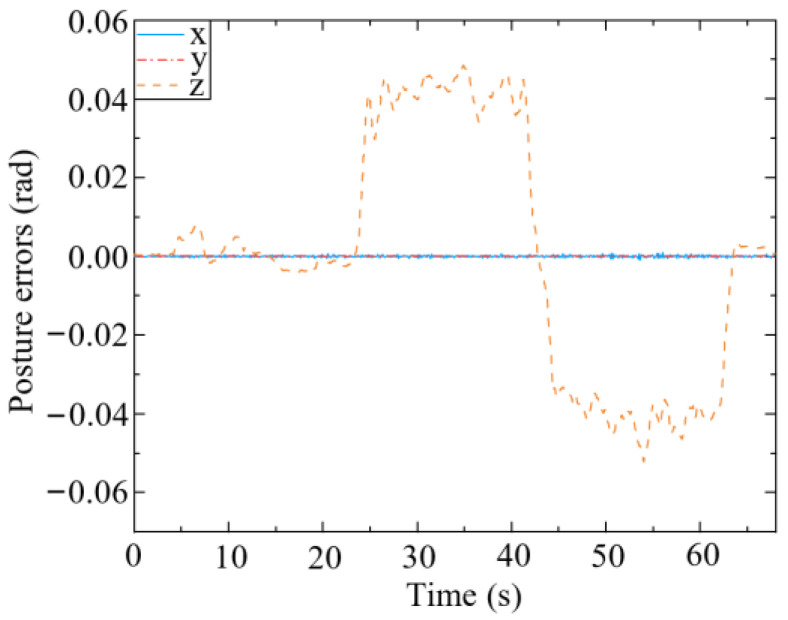
Posture errors of the end-effector relative to the mobile platform.

**Figure 23 biomimetics-09-00104-f023:**
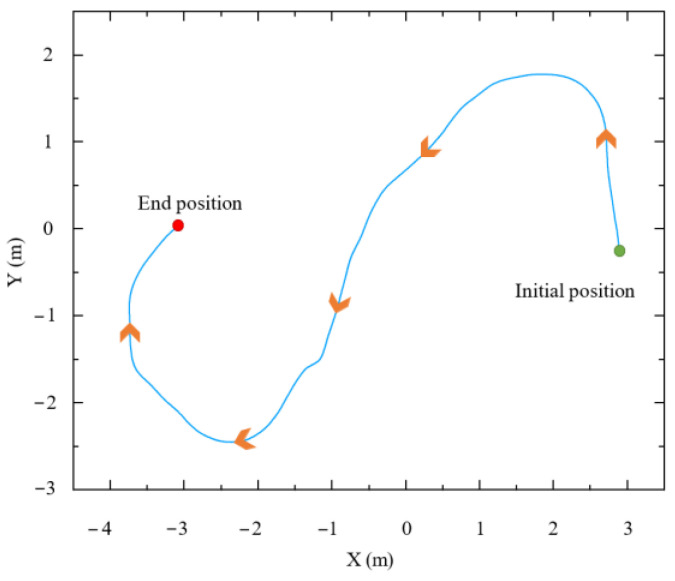
Reference path during assisted walking.

**Figure 24 biomimetics-09-00104-f024:**
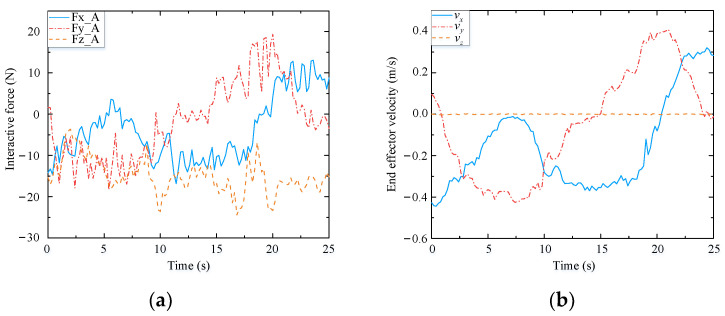
Test results of volunteer A. (**a**) Interactive forces; (**b**) End effector velocity.

**Figure 25 biomimetics-09-00104-f025:**
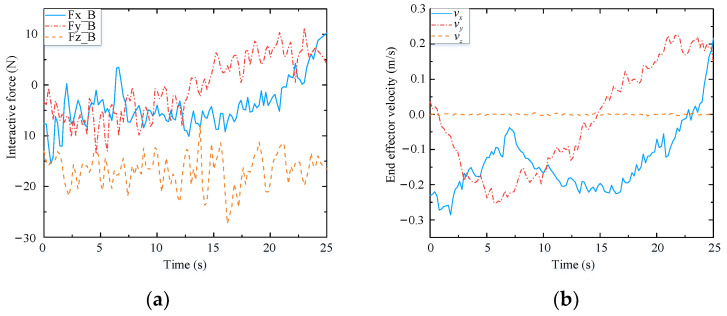
Test results of volunteer B. (**a**) Interactive forces; (**b**) End effector velocity.

**Figure 26 biomimetics-09-00104-f026:**
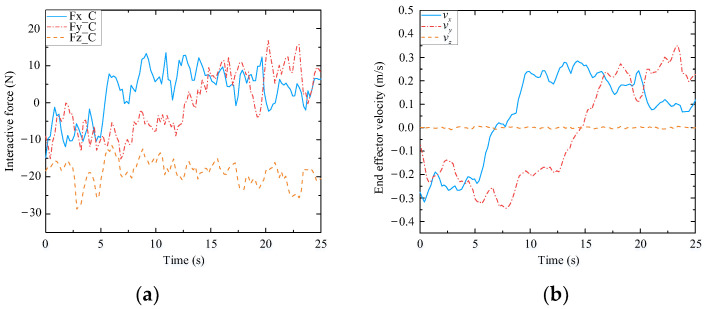
Test results of volunteer C. (**a**) Interactive forces; (**b**) End effector velocity.

**Table 1 biomimetics-09-00104-t001:** Fuzzy rule table.

*C* _v_		*F* _h_						
		NB	NM	NS	ZO	PS	PM	PB
*v* _h_	NB	VS	VS	S	M	B	VB	VB
	NM	VS	S	S	M	B	VB	VB
	NS	S	S	M	B	VB	B	B
	ZO	M	M	B	VB	B	M	M
	PS	B	B	VB	B	M	S	S
	PM	VB	VB	B	M	S	S	VS
	PB	VB	VB	B	M	S	VS	VS

**Table 2 biomimetics-09-00104-t002:** Control algorithm parameters and geometric parameters of moving manipulator.

Parameters	Values	Parameters	Values
(*x*_b_, *y*_b_, *φ*)	(0.3212 m, 0, 0)	*B* _z_	500 Ns/m
*β*	1.25	*K* _z_	1500 N/m
*ε*	0.2	*W* _lim_	diag{0.025, 0.025, 0.05, 0.1, 0.1, 0.1}
*μ*	0.1	*w*_1_, *w*_2_, *w*_3_	0.1, 0.8, 0.1
*C* _0_	50 Ns/m	*α*_1_, *α*_2_, *α*_3_	40, 1/1600, 1/200
*s*_a_, *s*_b_	10.3, 3.606	*K* _v_	diag{1, 1, 1}
*M* _z_	500 Ns^2^/m	*K* _w_	diag{1, 1, 10}
*L*	930 mm	*l* _0_	740 mm
*W*	699 mm	*w* _0_	349.5 mm
*H*	349 mm	*h* _0_	100 mm
*l* _1_	648 mm	*l* _2_	563 mm

**Table 3 biomimetics-09-00104-t003:** Specific statistics of the interaction force and end effector velocity.

Statistic		I	II	III	IV
||*F_h_*||(N)	maximum	25.70	23.88	22.14	20.85
	average value	10.61	10.81	9.61	8.52
||*v_e_*||(m/s)	maximum	0.2035	0.2290	0.2488	0.2373
	average value	0.1036	0.1112	0.1146	0.1126

**Table 4 biomimetics-09-00104-t004:** Specific joint torque data.

Optimization and Load		Joint Torque *τ*_m_ (Nm)	||*τ*_m_|| (Nm)
Before optimization	No load	[−0.1009, 1.6417, 1.0701, 0.7636, −0.2169, −0.0280]^T^	2.1169
	With load	[−0.1964, 2.0757, 1.1780, 0.6482, −0.1344, −0.0353]^T^	2.4848
After optimization	No load	[−0.0565, 0.8080, 0.7693, 0.3435, −0.1486, −0.0188]^T^	1.1782
	With load	[−0.0702, 0.7665, 1.0110, 0.6811, −0.1168, −0.0333]^T^	1.4468

**Table 5 biomimetics-09-00104-t005:** Performance comparison of volunteers A, B, and C.

Experimenters	A	B	C
Maximum of |*F*_x_| (N)	16.82	15.48	13.51
Maximum of |*F*_y_| (N)	18.30	13.63	16.79
Maximum of |*F*_z_| (N)	24.41	27.32	28.69
Maximum of |*v*_x_| (m/s)	0.45	0.37	0.32
Maximum of |*v*_y_| (m/s)	0.43	0.25	0.35
Maximum of |*v*_z_| (m/s)	0.0051	0.0093	0.0096

**Table 6 biomimetics-09-00104-t006:** Evaluation of volunteers A, B, and C.

Data	A	B	C
Comfort	8	7	9
Stability	8	9	8
Compliance	8	8	9
Average score	8	8	8.6

## Data Availability

Data is not available due to privacy and ethical restrictions.

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
