# Peer review of "Compliant-Control-Based Assisted Walking with Mobile Manipulator"

_biomimetics, 2024, doi:10.3390/biomimetics9020104_

Round 1
Reviewer 1 Report
Comments and Suggestions for Authors
This paper proposes a novel approach to assist human walking by using a mobile manipulator that imitates an experienced nurse. The main contributions of the paper are:
- A variable admittance controller that can adapt the virtual mass and damping parameters based on the fuzzy theory and the interaction forces and velocities.
- A velocity feedforward compensator that can reduce the inertia resistance of the manipulator by using a state observer to estimate the joint torque due to the external force.
- A null-space controller that can optimize the configuration of the mobile manipulator by using the projected gradient method and considering the singularity, force capacity, and gravity-induced deformation.
The paper is well-written and organized, and the proposed methods are clearly explained and illustrated. The paper also provides a suitable literature review and a detailed mathematical derivation of the kinematics and dynamics of the mobile manipulator. The paper validates the feasibility of the proposed methods by conducting a series of experiments involving different human volunteers.
However, the paper also has some limitations and areas for improvement, such as:
- The paper does not provide a clear definition and motivation of the assisted walking task and the criteria for evaluating the performance of the mobile manipulator. For example, what are the specific goals and challenges of assisted walking? How to measure the compliance, stability, and comfort of the human–robot interaction? How to compare the proposed methods with other existing methods or baselines?
- The paper states that the control of the robot is based on an “experienced nurse”, however there is not data or methods that seems to be collected from the nurse to drive the develop of this system. There are heuristic methods described when implementing the fuzzy model (line 235) but it is not clear where these came from. Did this come from nurse guidance?
- The paper does not discuss the potential ethical, social, and safety issues of using a mobile manipulator to assist human walking, especially for the elderly or disabled people. For example, how to ensure the reliability and robustness of the mobile manipulator in different scenarios and environments? I think this is worth at least a mention in the discussion.
- The paper does not provide sufficient details and analysis of the experimental. For example, when it comes to the effectiveness of the feedforward velocity control, a side-by-side comparison with the system not utilising the feedforward control is missing. This seems like an obvious comparison to evaluate the effectiveness of this contribution. Without this, the analysis seems to only validate the ability of the State Observer to measure the joint torques, rather than the benefit of the feedforward velocity. The RMS error for the first 3 joints is mentioned, but without a comparison of the error when not using the feedforward control, it is difficult to judge.
- Some of the design choices are not well explained. Why was a fuzzy controller utilised? How is the end effector stiffness K_opt determined?
- The need for the feedforward velocity commands was not explained in detail. I would assume that a manipulator under closed-loop control could reject external disturbances. Was this not the case for the UR3? Also, I assume this feedforward velocity technique is only relevant for manipulators that are velocity controlled? If torque controlled, I would think the joint torque due to the external force could be directly compensated as a feedforward torque. Some more detail on this would be great.
- Line 741: “Thus, it can be concluded that the proposed controller (48) can generate an appropriate velocity depending to the interaction forces applied by different humans and compliantly track the human user’s motion, meaning that the proposed approach is robust for the different gaits of volunteers.” I am not convinced that the performance of the system can be effectively captured through the max force/velocity numbers alone. Was any kind of user feedback sought? To me, this would be the more important metric of all.
***GENERAL IMPROVEMENTS***
There are also some small parts that could be improved, the following are some general comments or areas of ambiguity:
Be explicit about some of the variables. E.g. in line 152 make it clear that u_b = [v_b, omega_b]^T.
Line 163: “manipulator’s DOF is r”, how is ‘r’ used? How is it different from ‘m’? Shouldn’t ‘r’ be the DOF of the task space?
Is it possible to have a plot of Eq (6)? It would make interpreting the equation easier for the reader.
In Eq(6), what is C_0? I assume it is detailed in [31] but for completeness it should be included here.
I do not understand the notation in Eq(7). Is C_vi a vector? I thought it was a scalar with respect to dof ‘I’.
Transition from line 408 to 409 is awkward. It seems like something is missing there.
Equation (3) is formatted poorly, it could be easily split over several lines. Same with Eq(6). I assume this is an issue with the pre-print.
Figure 14: make it clearer what the little sub-plot represents.
***ERRORS***
Line 102-103: “… to accurately to track …”
Line 147: “j-th(j = 1, 2, 3, 4, 5, 6)” space between is missing.
Line 344: “… and [] is the control period” I think ‘T’ is missing.
In (38) should the ‘i' subscripts actually be ‘j’?
Line 601: “position error in the [] direction is the greatest” missing character?
Comments on the Quality of English Language
The quality of English is acceptable, however several formatting mistakes were found.
Line 102-103: “… to accurately to track …”
Line 147: “j-th(j = 1, 2, 3, 4, 5, 6)” space between is missing.
Line 344: “… and [] is the control period” I think ‘T’ is missing.
In (38) should the ‘i' subscripts actually be ‘j’?
Line 601: “position error in the [] direction is the greatest” missing character?
Reviewer 2 Report
Comments and Suggestions for Authors
1. A control system for the mobile robot-manipulator unit, assisting patients in walking is presented. Although the structure of the control system as well as subsequent control algorithms are quite complicated, yet they are well-described and clearly explained in the text. The most important parts of the paper to understand the structure of the proposed control system are Fig. 4, together with the equations shown in it and explained in subsequent places of the paper.
2. Geometric parameters as well as other parameters of the considered mobile robot-manipulator unit could be included in the paper to let the interesting Readers reproduce some of the Authors findings.
3. Hardware and software implementation problems of the the proposed structure of the control system could be explained: Is there the global controller for the whole unit or are there separate controllers for the mobile robot and for the manipulator? If separate controllers, then how does the mobile controller and the manipulator controller communicate with each other? What is the computation performance (computation time) of the whole control loop? What software is used to implement the control algorithms? What sensors and actuators of the mobile platform are used? What sensors of the manipulator are used? etc.
4. It is not clear if the six-axis force sensor shown in Fig. 5 is used in the control system shown in Fig. 4 or is it used only to verify torque estimates as shown in Fig. 13. If not then which sensors are used to verify the estimates in Fig. 5?
5. It seems that the manipulator pose optimization with the objective function Eq. (38) and the null-space controller Eq. (40) is performed only once, before the assisted walking, and then during the walking this optimized pose is maintained, i.e. the null-space controller is switched off then - is this the right observation?
6. Although the amplitudes of the interactive forces as shown in Figs. 21-23 are small enough, yet it seems that for the patient assisted with the proposed mobile robot-manipulator unit uncomfortable experiences might come from the oscillations of these forces. Such oscillations might be a serious deficiency of the developed control algorithms - with such oscillations the patient might be afraid about the robustness/rigidity of this intelligent support.
7. Line 223: The definition of $C_{vi}$ in Eq. (7) is imprecise: for which $F_{hi}$ $C_{vi}=C_{min}$ and for which $F_{hi}$ $C_{vi}=C_{max}$?
8. Line 344: “... and ??? is the control period.” - Include the symbol for the “control period”.
9. Line 601: “... position error in the ??? direction is the greatest...” - Include $\Delta x$ for the “position error”.
Reviewer 3 Report
Comments and Suggestions for Authors
please see the attachment.

Round 2
Reviewer 2 Report
Comments and Suggestions for Authors
The Authors have addressed all my previous comments and introduced corresponding corrections in the text. The paper has been improved and may be accepted for publication. The detailed comments to the Authors' responses are given below.
-------------------------------------
Previous comment 2. Geometric parameters as well as other parameters of the considered mobile robot-manipulator unit could be included ....
New comment: Thank you for explaining the geometric parameters and their values in the revised manuscript.
-------------------------------------
Previous comment 3. ... implementation problems of the the proposed structure of the control system could be explained:...
New comment: Thank you for explaining hardware and software structures of the proposed robot controller in the revised manuscript.
-------------------------------------
Previous comment 4. It is not clear if the six-axis force sensor ... is used in the control system ...
New comment: Thank you for the explanation about the six-axis force sensor in the revised manuscript.
-------------------------------------
Previous comment 5. It seems that the manipulator pose optimization ... is performed only once ...
New comment: Thank you for the additional explanation about the manipulator pose optimization in the revised manuscript.
-------------------------------------
Previous comment 6. ... the amplitudes of the interactive forces ... uncomfortable experiences might come from the oscillations of these forces ...
New comment: I am happy the Authors agree with the suggestion the oscillations of the interactive forces might be uncomfortable for a few patients. I hope the Authors will be able to perform the planned research on the amplitude control of the interactive forces in the future.
-------------------------------------
New comment to previous comments 7, 8, 9: Thank you for these corrections to the formulas and the descriptions of formulas.
